# FROST: FILTERING REASONING OUTLIERS WITH ATTENTION FOR EFFICIENT REASONING

**Haozheng Luo**[†♮*]  **Zhuolin Jiang**[♮]  **Md Zahid Hasan**[♮‡*]  **Yan Chen**[†]  **Soumalya Sarkar**[♮]

[†]Department of Computer Science, Northwestern University, Evanston, IL 60208 USA
[♮]RTX Technology Research Center (RTRC), East Hartford, CT 06118 USA
[‡]Department of Electrical and Computer Engineering, Iowa State University, Ames, IA 50011 USA

hluo@u.northwestern.edu    {zhuolin.jiang,soumalya.sarkar}@rtx.com
zahid@iastate.edu    ychen@northwestern.edu

## ABSTRACT

We propose **FROST**, an attention-aware method for efficient reasoning. Unlike traditional approaches, FROST leverages attention weights to prune uncritical reasoning paths, yielding shorter and more reliable reasoning trajectories. Methodologically, we introduce the concept of reasoning outliers and design an attention-based mechanism to remove them. Theoretically, FROST preserves and enhances the model's reasoning capacity while eliminating outliers at the sentence level. Empirically, we validate FROST on four benchmarks using two strong reasoning models (Phi-4-Reasoning and GPT-oss-20B), outperforming state-of-the-art methods such as TALE and ThinkLess. Notably, FROST achieves an average **69.68%** reduction in token usage and a **26.70%** improvement in accuracy over the base model. Furthermore, in evaluations of attention outlier metrics, FROST reduces the maximum infinity norm $\|\mathbf{x}\|_\infty$ by **15.97%** and the average kurtosis by **91.09%** compared to the base model.

## 1 INTRODUCTION

We observe that large reasoning models (LRMs) often generate numerous irrelevant steps, which we term **reasoning outliers**. To mitigate this, we introduce **FROST**, an efficient reasoning method that leverages attention weights to prune uncritical reasoning paths, producing shorter and more reliable trajectories. More specifically, FROST replaces the standard $\mathrm{Softmax}$ function with $\mathrm{Softmax}_1$, enabling attention to better identify and suppress outliers. This directs LRMs toward critical reasoning steps, thereby enhancing their overall reasoning capacity.

Efficient reasoning is critical for large reasoning models (LRMs), which have shown strong performance in tasks such as mathematical problem-solving (Luo et al., 2025a; Yang et al., 2024; Shao et al., 2024), coding (Ding et al., 2024a;b), and scientific question answering (Comanici et al., 2025; Hurst et al., 2024). Yet, these models often generate large amounts of uncritical information—commonly arising from redundant self-verification—that introduce inefficiencies and potential inaccuracies. Numerous methods have been proposed to improve reasoning efficiency. Token-level approaches such as TALE (Han et al., 2025) and R2R (Fu et al., 2025) risk pruning essential reasoning steps, as reasoning paths are naturally sentence-based. Sentence-level approaches, including DRP (Jiang et al., 2025b) and GRPO-S (Tan & Pan, 2025), perform iterative refinement of reasoning paths, but this often comes at the cost of increased computational cost and latency.

To address these challenges, we propose **FROST**, a reasoning method that improves efficiency by pruning uncritical reasoning paths through attention weights. We observe that LRMs typically assign low attention to uncritical steps and higher attention to critical ones, consistent with findings that critical steps exhibit higher sentence entropy (Tan & Pan, 2025). We therefore introduce the concept of *reasoning outliers*—uncritical steps with both low attention weights and low entropy (Wang et al., 2025; Fu et al., 2025)—and design FROST to eliminate them, yielding shorter and more reliable reasoning paths. Our approach sharpens the attention distribution of LRMs, suppressing

---

*This work was carried out at the RTX Technology Research Center during a Summer 2025 internship.

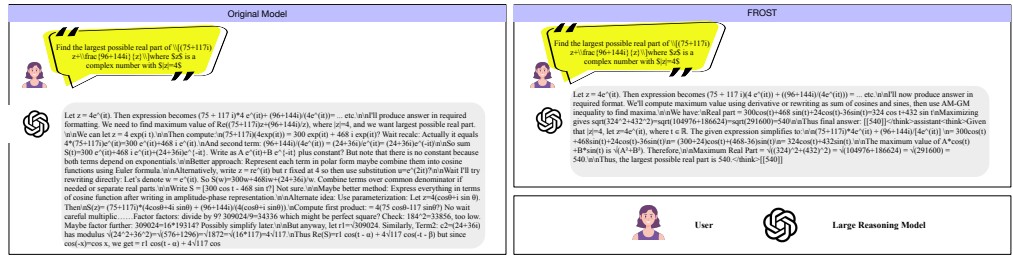

Figure 1: **The Example of The GPT-OSS-20B Model**.

low-weight steps while preserving high-weight ones. Building on prior work (Luo et al., 2025b; Hu et al., 2024; Xiao et al., 2024), we adopt $\text{Softmax}_1$ in place of $\text{Softmax}$, which effectively drives low weights to zero while maintaining large weights. Finally, we propose a training strategy that integrates $\text{Softmax}_1$ with supervised fine-tuning on reasoning tasks, producing efficient reasoning models without sacrificing accuracy.

**Contributions.** We present **FROST** (as shown in Figure 1), a reasoning outlier–free LRM designed to enhance reasoning efficiency. Our main contributions are:

- We introduce the concept of **reasoning outliers** and propose **FROST** to prune uncritical reasoning steps characterized by low attention.

- Theoretically, we analyze $\text{Softmax}_1$ and show its effectiveness in suppressing low attention weights while preserving high ones, thereby enhancing the reasoning capacity of LRMs.

- Methodologically, we design a training strategy that combines $\text{Softmax}_1$ with supervised fine-tuning, enabling efficient reasoning without sacrificing accuracy.

- Empirically, we demonstrate the effectiveness of FROST across multiple benchmarks, achieving up to a **26.70%** accuracy gain while reducing reasoning path length by **69.68%** compared with base models. We also measure attention outlier values to verify their impact on efficient reasoning: FROST reduces the maximum infinity norm $\|\mathbf{x}\|_\infty$ by **15.97%** and the average kurtosis by **91.09%**. In addition, FROST cuts inference time by at least 28.6% and reduces training time by 42.2% relative to other SFT baselines.

## 2 RELATED WORK

**Reasoning Models.** In recent years, Large Language Models (LLMs) such as DeepSeek-R1 (Guo et al., 2025), OpenAI o1 (Jaech et al., 2024), and Gemini 2.0 Pro (Team et al., 2023) have demonstrated strong reasoning capabilities, particularly on mathematical and logical tasks (Hao et al., 2024). To further improve reasoning performance, numerous methods are proposed, falling into the main paradigms (Ke et al.): inference scaling and learning-to-reason. For inference-time scaling, numerous methods have been proposed, including few-shot prompting (Brown et al., 2020), in-context learning (Brown et al., 2020), Chain-of-Thought (CoT) reasoning (Wei et al., 2022), and Search & Planning (SP) (Besta et al., 2024). Numerous studies focus on improving the LLM reasoning at inference time, with CoT emerging as a key technique. CoT strengthens the model's reasoning process and generates interpretable reasoning traces. A simple example involves adding a prompt like "Let's think step by step" after a question (Wei et al., 2022). Recent research increasingly combines CoT with other inference-time scaling methods, such as ReAct (Yao et al., 2023), Self-Ask (Press et al., 2023) and agentic reasoning (Pan et al., 2025; 2024), to further enhance reasoning capabilities. For learning-to-reason approaches, many methods aim to build reasoning ability through alignment, including reinforcement learning (RLHF (Ouyang et al., 2022), DPO (Rafailov et al., 2023), GRPO (Ramesh et al., 2024)), supervised fine-tuning, and energy-based model (EBM) reasoners (Jiang et al., 2025a). However, LLMs with reasoning capabilities—particularly those with smaller parameter sizes—often generate excessively detailed reasoning chains, including unnecessary tracebacks and redundant alternative paths (Hou et al., 2026; Chen et al., 2025). This overthinking not only increases computational cost during inference but can also negatively impact response quality on accuracy (Cuadron et al., 2025) and safety (Kumar et al., 2025). To address this, we propose an attention-aware adaptation method that optimizes reasoning paths, yielding efficient reasoning models.

**Efficient Reasoning Methods.**   To address overthinking, current approaches to optimizing reasoning paths fall into three categories (Sui et al., 2025): prompt-based methods, supervised fine-tuning, and reinforcement learning. Prompt-based methods (Liu et al., 2025; Xu et al., 2025a; Han et al., 2025) introduce token-budget constraints to shorten reasoning paths. For instance, TALE (Han et al., 2025) limits the token budget per instance to reduce reasoning length while maintaining task accuracy. Supervised fine-tuning (SFT) methods (Ma et al., 2025; Xia et al., 2025a) improve reasoning conciseness by training models on compressed reasoning paths. For example, DRP (Jiang et al., 2025b) fine-tunes models on distilled reasoning data by pruning unrelated reasoning steps. Reinforcement learning (RL) methods (Hou et al., 2026; Li et al., 2025; Yi et al., 2025) guide concise reasoning by introducing reward functions that penalize overly long reasoning paths. For example, Chia et al. (2024) introduce a reward score based on reference loss and exploration loss from diverse paths, encouraging favorable reasoning branches and penalizing unfavorable ones to improve overall problem-solving performance. However, prompt-based methods rely on handcrafted prompts and often perform unreliably on complex problems. In contrast, SFT and RL approaches require substantial computational resources for fine-tuning, limiting accessibility for users without adequate hardware. To address these challenges, we propose a new reasoning outlier–removal strategy that eliminates reasoning outliers through attention analysis. Recent studies (Choi et al., 2025; Cai et al.) also analyze internal attention patterns in reasoning models, particularly at the sentence level, but their objectives differ substantially from ours and focus on KV-cache–based inference efficiency. Think Clearly (Choi et al., 2025) examines sentence-level attention spikes near the end-of-thinking token and uses these patterns to prune redundant sentences for faster decoding. In contrast, our Figure 3 analyzes sentence-level contributions to the final-answer token, enabling attribution of which specific reasoning sentences actually affect the model's prediction, rather than identifying redundancy for pruning. R-KV (Cai et al.) likewise detects redundant attention interactions to compress the KV cache, but does not study how individual reasoning steps functionally influence final-answer formation. Our work therefore provides a finer-grained, component-level attribution analysis of the reasoning trace—going beyond redundancy detection to clarify how different reasoning segments vary in contribution, which constitutes the key novelty relative to these approaches.

## 3   REASONING OUTLIER

In this section, we analyze the attention distribution of reasoning traces generated by LRMs. We then examine the impact of different components of the trace on final answer prediction, followed by our definition and characterization of reasoning outliers.

### 3.1   ATTENTION DISTRIBUTION OF REASONING TRACES

We consider representative LRMs, including DeepSeek-R1(Guo et al., 2025), Phi-4 (Abdin et al., 2025), and GPT-4o (Hurst et al., 2024), which generate text in an autoregressive manner by predicting the next token given the preceding context. To study the attention distribution, we visualize the attention heatmap of each token in the reasoning trace when predicting the final answer.

Let the reasoning process be a sequence of tokens $T = [t_1, t_2, \ldots, t_n]$, where each $t_i$ denotes a token in the process. The attention weight matrix $A$ is defined as:

$$A = [a_{ij}] \quad \text{where} \quad a_{ij} = \text{AttentionWeight}(t_i, t_j).$$

Here, $a_{ij}$ represents the attention weight from token $t_i$ to token $t_j$.

As an illustrative example, we use a sample question from GSM8K (Cobbe et al., 2021) and generate the reasoning trace with the Phi-4-Reasoning model (Abdin et al., 2025). Figure 2 shows the corresponding attention heatmap. The results indicate that in the shallow layers, the attention distribution is relatively uniform across all tokens. However, as we move to deeper layers and later heads, the model begins to focus more on specific tokens, particularly those in the reasoning steps and the final answer. This suggests that the model progressively refines its focus towards the most relevant parts of the reasoning trace as it processes the information.

### 3.2   IMPACT OF REASONING TRACE COMPONENTS ON ANSWER PREDICTION

To quantify the impact of different components of the reasoning trace on final answer prediction, we conduct an additional experiment analyzing the summed attention weight distribution to the final answer token `</think>`, which allows us to measure how strongly each reasoning step contributes to the model's ultimate decision and provides insights into whether the model grounds its prediction

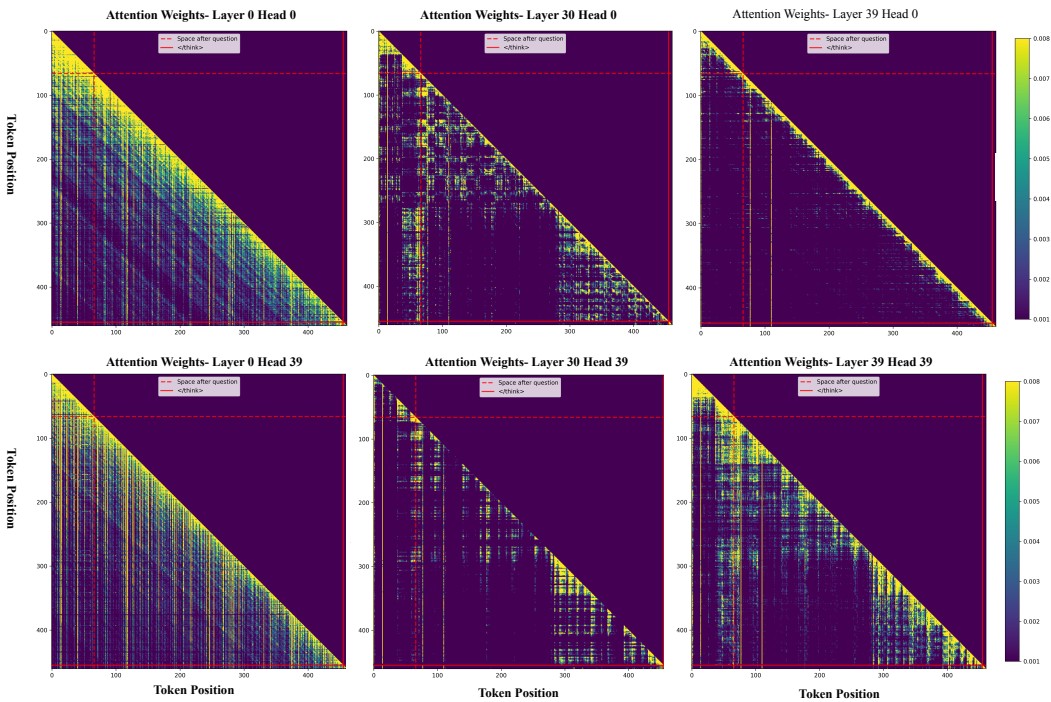

Figure 2: **Attention Heatmap of Reasoning Tokens.** We use the Phi-4-Reasoning model (Abdin et al., 2025) to generate a reasoning trace for a sample GSM8K question (Cobbe et al., 2021). The figure shows attention heatmaps from transformer layers 0, 30 and 39, with the first head (top row) and last head (bottom row). Yellow indicates higher attention weights and blue indicates lower ones. In shallow layers, contributions to the final answer are nearly uniform, while deeper layers and later heads highlight specific tokens with stronger influence.

in meaningful intermediate reasoning or relies on superficial correlations. We divide the reasoning process into four components: the question $Q$, the reasoning steps $R_1, R_2, \ldots, R_m$, and the final answer $A$. For each component, we compute the total attention weight contributing to the first token of the final answer: $W_{\text{trace}} = \sum_{t_i \in T_{\text{trace}}} a_{iA}$, where $T_{\text{trace}}$ is the set of tokens in a given component, and $a_{iA}$ denotes the attention weight from token $t_i$ to the `</think>` token.

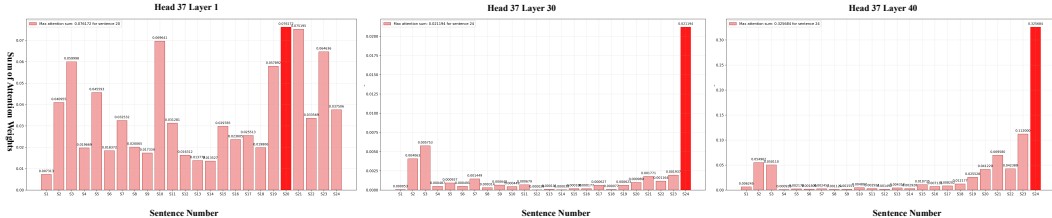

Figure 3: **Total attention weight distribution to the final answer token `</think>` from different components of the reasoning trace.** We visualize the total attention weight distribution of the Phi-4-Reasoning model on a sample GSM8K question, using transformer layers 1, 30, and 40. The results show that a few reasoning traces contribute strongly to the final token `</think>`, while many traces have negligible influence, particularly in the layers 30 and 40.

As shown in Figure 3, different reasoning traces contribute unequally to final answer generation. While a few traces show strong influence, most contribute weakly, and some exhibit almost no contribution at all.

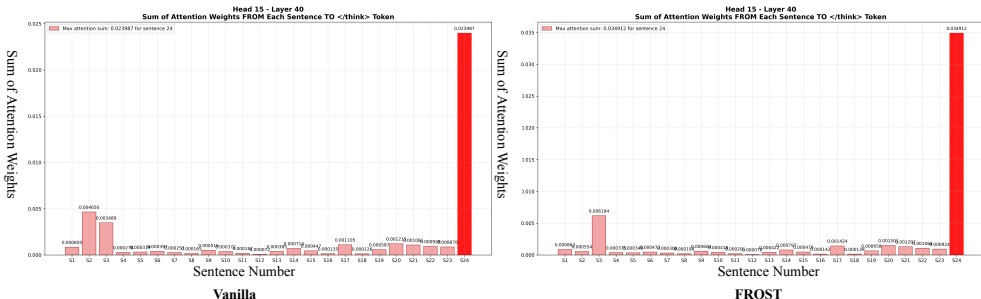

Figure 4: **Theoretical Analysis of Reasoning Outlier Removal.** We conduct a theoretical analysis with Phi-4-Reasoning model to demonstrate that removing reasoning outliers using the $\text{Softmax}_1$ function (FROST) can preserve or even enhance the model's reasoning capacity. As shown in the figure, the attention weight distribution before and after outlier removal indicates that the model's focus on critical reasoning traces is maintained or improved, while the influence of outliers is significantly reduced.

### 3.3 DEFINING AND CHARACTERIZING REASONING OUTLIERS

As observed in Section 3.1, many reasoning traces contribute negligibly to the final answer. These traces often correspond to verification, self-checking, or repetition of prior reasoning steps. Their presence forces LRMs to generate more tokens than necessary, substantially reducing reasoning efficiency. A potential cause (Sui et al., 2025) is that model developers often encourage extended reasoning steps to maximize accuracy. In the meantime, the model may generate redundant or irrelevant information, leading to inefficient and incorrect reasoning. As a result, we define reasoning traces with low attention weight and negligible contribution to the final answer as **reasoning outliers**.

To identify and remove reasoning outliers, we observe that they share similar characteristics with attention outliers (Luo et al., 2025b; Hu et al., 2024). Motivated by this, we adopt $\text{Softmax}_1$ (eq. (3.1)) to detect and eliminate reasoning outliers during the reasoning process, and provide a comprehensive proof of its efficiency in Section 5.

$$\text{Softmax}_1(x_i) = \frac{\exp(x_i)}{\sum_j \exp(x_j) + 1}, \tag{3.1}$$

where $x_i$ represents the attention weight of token $t_i$.

**Theoretical Analysis.** We conduct a theoretical analysis to show that removing reasoning outliers with the $\text{Softmax}_1$ function preserves, and can even enhance, the reasoning capacity of LRMs. In our experiments, we use the Phi-4-Reasoning (Abdin et al., 2025) to generate reasoning traces for a sample GSM8K question (Cobbe et al., 2021). Specifically, we compare the last layer's attention distribution in head 15 under vanilla attention and $\text{Softmax}_1$ attention (FROST). As shown in Figure 4, $\text{Softmax}_1$ reduces the influence of outliers while maintaining or strengthening focus on critical reasoning traces. This analysis supports our approach of using $\text{Softmax}_1$ to effectively identify and eliminate reasoning outliers, thereby improving the efficiency and reliability of LRMs. For more details of the theoretical proof, please refer to Section 5.

## 4 FROST

To enhance the reasoning efficiency of LRMs, we propose supervised fine-tuning (SFT) with reasoning outlier removal, as illustrated in Figure 5.

In the SFT stage, we train on math problems with detailed reasoning steps and answers. During training, we replace the vanilla $\text{Softmax}$ with $\text{Softmax}_1$ (eq. (3.1)), enabling the model to focus on critical reasoning traces while suppressing outliers. Unlike prior methods that employ $\text{Softmax}_1$ for outlier removal—requiring either training from scratch (Hu et al., 2024) or multi-step continual learning (Luo et al., 2025b)—our approach achieves effective outlier removal with only a few steps of fine-tuning from existing pretrained checkpoints, making it more efficient and practical. We optimize model parameters using cross-entropy loss and apply LoRA (Hu et al., 2021) to further reduce training cost.

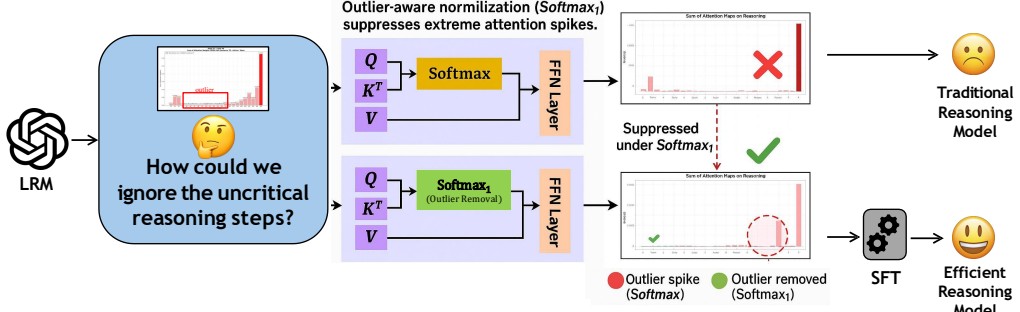

Figure 5: **Overview of the** FROST **workflow** We replace the vanilla $\mathrm{Softmax}$ layer with an outlier-removal layer based on $\mathrm{Softmax}_1$, followed by SFT to adapt model parameters to the new activation function. We observe that our method significantly reduces the number of low-attention sentences.

## 5 THEORETICAL ANALYSIS

In this section, we provide a brief theoretical analysis showing that $\mathrm{Softmax}_1$ can operate at the sentence level to remove reasoning outliers in LRMs. We provide a theoretical proof that our method achieves deployment-time suppression in efficient reasoning, consistent with our findings in Figure 4.

**Setup.** Let a token sequence be partitioned into sentences $\{S_i\}_{i=1}^m$. For a query $q \in \mathbb{R}^d$ and keys $\{k_t\} \subset \mathbb{R}^d$, define token compatibilities $z_t = \mathrm{Softmax}_1\big(\frac{\langle q, k_t \rangle}{\sqrt{d}}\big) v_t$, where $t$ is the token index in $S_i$ and $v_t \in \mathbb{R}^d$ denotes the token value for each token in sentence $S_i$. Let $\phi : \mathbb{R}^{|S_i|} \to \mathbb{R}$ be a *monotone* pooling operator (e.g., sum/mean/logsumexp/max). Define sentence scores $s_i = \phi\big(\{z_t\}_{t \in S_i}\big)$ and $s = (s_1, \ldots, s_m) \in \mathbb{R}^m$. Define the probability simplex $\Delta^{m-1} = \Big\{ \alpha \in \mathbb{R}^m \,\big|\, \alpha_i \geq 0, \sum_{i=1}^m \alpha_i = 1 \Big\}$.

**Assumption 5.1** (Softmax$_1$ operator). There exists a $\mathrm{Softmax}_1$ mapping $\sigma_1 : \mathbb{R}^m \to \Delta^{m-1}$ such that:

1. **Order preservation:** If $x_i \geq x_j$ then $\sigma_1(x)_i \geq \sigma_1(x)_j$.
2. **Shift invariance:** $\sigma_1(x + c\mathbf{1}) = \sigma_1(x)$ for all $c \in \mathbb{R}$.
3. **Tail contraction:** There exists $\kappa \in (0, 1)$ such that for all $x \in \mathbb{R}^m$, $\frac{\|\sigma_1(x)\|_\infty}{\mathrm{median}(\sigma_1(x))} \leq \kappa \frac{\|x\|_\infty}{\mathrm{median}(x)}$.
4. **Smoothness and positivity:** $\sigma_1$ is continuously differentiable on $\mathbb{R}^m$ and $\sigma_1(x)_i > 0$ for all finite $x$.

We write the sentence-level attention as $\alpha = \sigma_1(s) \in \Delta^{m-1}$ and the layer output as $y = \sum_{i=1}^m \alpha_i v_i$, which $\alpha_i$ and $v_i$ are attention probabilities and token values corresponding to sentence $s_i$. Assume $\|v_i\|_\infty \leq B_v$ and that all linear maps used below have finite operator norms $B_v$, which is a constant.

**Lemma 5.1** (Monotone pooling preserves sentence dominance). If $\phi$ is monotone coordinatewise, then for any $i, j$, $\big(\forall t \in S_i, \exists\, t' \in S_j : z_t \geq z_{t'}\big) \implies s_i \geq s_j$. Consequently, by Assumption 5.1(P1), $\alpha_i = \sigma_1(s)_i \geq \sigma_1(s)_j = \alpha_j$.

*Proof.* See Appendix C.1 for a detailed proof. □

**Theorem 5.1** (Softmax$_1$ suppresses sentence-level attention outliers). Let $s = (s_1, \ldots, s_m)$ be the sentence scores built via a monotone pooling $\phi$. If $s$ is heavy-tailed (e.g., $\|s\|_\infty/\mathrm{median}(s) \gg 1$), then for $\alpha = \sigma_1(s)$

$$\frac{\|\alpha\|_\infty}{\mathrm{median}(\alpha)} \leq \kappa \cdot \frac{\|s\|_\infty}{\mathrm{median}(s)} \qquad \text{for some } \kappa \in (0, 1), \tag{5.1}$$

so the relative dominance of outliers contracts at the sentence level.

*Proof.* See Appendix C.2 for a detailed proof. □

**Theorem 5.2** (Deployment-time suppression of low-attention sentences). Let the output logits be $\ell = W_o y$ with $\|W_o\|_{op} =: B_o$ and $\|v_i\| \leq B_v$. For a sentence $i$ with $\alpha_i \leq \varepsilon$, its one-layer contribution to logits is bounded by

$$\|\Delta \ell_i\| = \|W_o(\alpha_i v_i)\| \leq B_o \varepsilon \|v_i\| \leq B_o B_v \varepsilon. \tag{5.2}$$

For $L$ stacked layers with Jacobians $\{J_\ell\}_{\ell=1}^L$ and $\|J_\ell\|_{op} \leq B_\ell$,

$$\|\Delta \ell_i^{(L)}\| \leq \varepsilon \Big(\prod_{\ell=1}^L B_\ell\Big) B_v B_o. \tag{5.3}$$

Since the Softmax$_1$ map sm $: \mathbb{R}^V \to \Delta^{V-1}$ is 1-Lipschitz in the $\ell_\infty \to \ell_1$ norm,

$$\big\|\text{sm}(\ell + \Delta \ell_i^{(L)}) - \text{sm}(\ell)\big\|_1 \leq \|\Delta \ell_i^{(L)}\| \leq B_o B_v \Big(\prod_{\ell=1}^L B_\ell\Big) \varepsilon. \tag{5.4}$$

Let $B := \max_{\ell \in [L]} B_\ell$ be the largest operator norm over $L$ layers. Then

$$\big\|\text{sm}(\ell + \Delta \ell_i^{(L)}) - \text{sm}(\ell)\big\|_1 \leq B_o B_v B^L \varepsilon = O\big(B_o B_v B^L \varepsilon\big). \tag{5.5}$$

In practice $B_o, B_v, B$ are approximately constant, so the bound reduces to $O(\varepsilon)$. Therefore, low-attention sentences are effectively skipped at inference.

*Proof.* See Appendix C.3 for a detailed proof. □

# 6 EXPERIMENTAL STUDIES

We conduct a series of experiments to evaluate FROST in providing efficient reasoning, benchmarking its performance on GPT-oss (Agarwal et al., 2025), Magistral-Small-1.1 (Rastogi et al., 2025) and Phi-4-Reasoning (Abdin et al., 2025). Each evaluation is conducted three times with different random seeds, and we report the average and standard deviation for each metric.

**Models.** In our experiments, we use Phi-4-Reasoning (Abdin et al., 2025),Magistral-Small-1.1 (Rastogi et al., 2025) and GPT-oss (Agarwal et al., 2025) as backbone models for efficient reasoning. Specifically, we adopt the Phi-4-Reasoning[*], Magistral-Small-1.1[*] and GPT-oss-20B-finetune[*] checkpoints, both finetuned on mathematical datasets with detailed reasoning steps and answers using SFT under the FROST method.

**Datasets.** Following the setup in (Zhao et al., 2025a), we use OpenR1 (Hugging Face, 2025) as the training corpus. To evaluate reasoning efficiency and generalization on complex mathematical problems, we adopt four out-of-domain benchmarks: GSM8K (Cobbe et al., 2021), MATH500 (Lightman et al., 2024), AIME24 (of America, 2024), and Minerva (Dyer & Gur-Ari, 2022). All datasets are designed for mathematical question answering.

**Metrics.** To evaluate the effectiveness of our efficient reasoning strategy, we report pass@1 as the accuracy metric and use the number of tokens in the reasoning response to measure token efficiency.

**Baselines.** We select five representative methods covering key paradigms of efficient reasoning: **(1) TALE** (Han et al., 2025): a prompt-based approach that uses a soft token budget to generate concise reasoning responses. **(2) DRP** (Jiang et al., 2025b): an SFT-based method that distills reasoning paths from a teacher model and applies step-level pruning to produce concise, skill-aware reasoning traces. **(3) SelfBudgeter** (Li et al., 2025): a reinforcement learning-based method that

---

[*]https://huggingface.co/microsoft/Phi-4-reasoning
[*]https://huggingface.co/mistralai/Magistral-Small-2507
[*]https://huggingface.co/openai/gpt-oss-20b

Table 1: **Comparison of FROST with Efficient Reasoning Methods.** We evaluate reasoning path efficiency by comparing FROST against four baselines across four mathematical datasets (GSM8K, MATH500, AIME24, and Minerva). Pass@1 and token usage (#Tk) are reported as evaluation metrics, with variance omitted since it is consistently $\leq 2\%$. Best results are shown in **bold**, and second-best are underlined. In most settings, FROST achieves the best performance among all methods. Specifically, it improves accuracy by 26.70% while reducing token usage by 69.68% compared to the base model.

| Type | Method | GSM8K | | MATH500 | | AIME24 | | Minerva | | $\Delta_{\text{Pass@1}}$ | $\Delta_{\text{#Tk}}$ |
|---|---|---|---|---|---|---|---|---|---|---|---|
| | | Pass@1 | #Tk | Pass@1 | #Tk | Pass@1 | #Tk | Pass@1 | #Tk | | |
| Phi-4-Reasoning | Base | 0.9242 | 1017.70 | 0.5480 | 1721.95 | 0.0667 | 1017.70 | 0.2500 | 1898.86 | 0.000 | 0.00 |
| | TALE | **0.9500** | 1716.60 | 0.5800 | 1874.43 | 0.2900 | 2069.97 | 0.2627 | 2093.17 | **+0.074** | +524.49 |
| | DRP | 0.8340 | 721.00 | **0.6200** | 2122.00 | **0.3333** | 6135.00 | 0.2701 | 1289.50 | +0.067 | +1152.69 |
| | SelfBudgeter | 0.9189 | 1507.14 | 0.5347 | 1195.18 | 0.1342 | 1372.83 | 0.2357 | 2618.23 | +0.009 | +259.30 |
| | ThinkLess | 0.9279 | 1421.90 | 0.5414 | 1101.21 | 0.1608 | 1405.40 | 0.2575 | 1708.70 | +0.025 | -4.75 |
| | Ours | 0.9311 | **154.33** | 0.5980 | **344.37** | 0.2667 | **899.80** | **0.2716** | **401.19** | +0.070 | **-964.13** |
| GPT-OSS-20B | Base | 0.8704 | 1275.23 | 0.5400 | 1575.36 | 0.1333 | **1003.57** | 0.2574 | 1586.95 | 0.000 | 0.00 |
| | TALE | 0.8283 | 2664.41 | 0.5454 | 3878.87 | 0.2000 | 1354.67 | 0.2700 | 3262.47 | +0.011 | +1430.33 |
| | DRP | 0.7880 | 902.50 | **0.6146** | 4137.00 | **0.2245** | 4983.00 | 0.2715 | 1885.15 | +0.024 | +1616.64 |
| | SelfBudgeter | 0.8610 | 1850.00 | 0.5340 | 2285.00 | 0.1320 | 1256.00 | 0.2550 | 1298.00 | -0.005 | +312.47 |
| | ThinkLess | 0.8740 | 1785.00 | 0.5410 | 2206.00 | 0.1600 | 1205.00 | 0.2580 | 1220.00 | +0.008 | +244.22 |
| | Ours | **0.8764** | **377.17** | 0.5800 | **680.89** | 0.1667 | 1009.60 | **0.2794** | **691.71** | **+0.025** | **-669.94** |
| Magistral-Small-1.1 | Base | 0.6075 | 2664.41 | 0.1480 | 1389.89 | 0.0000 | 537.13 | 0.0699 | 1288.04 | 0.000 | 0.00 |
| | TALE | 0.7146 | 1516.86 | **0.3040** | 723.91 | 0.0333 | 967.43 | 0.1544 | 748.18 | **+0.095** | -480.77 |
| | DRP | 0.6500 | 902.50 | 0.2100 | 1680.33 | 0.0450 | 1350.77 | 0.1120 | 1604.22 | +0.048 | -85.41 |
| | SelfBudgeter | 0.6900 | 1850.00 | 0.2300 | 1520.00 | 0.0520 | 1256.00 | 0.1300 | 1298.00 | +0.069 | +11.13 |
| | ThinkLess | 0.7200 | 1785.00 | 0.2500 | 1405.00 | 0.0600 | 1205.00 | 0.1450 | 1220.00 | +0.087 | -66.12 |
| | Ours | **0.7551** | **137.55** | **0.3040** | **98.20** | **0.0974** | 149.93 | **0.1551** | **109.23** | **+0.122** | **-1346.14** |

iteratively shortens the reasoning path by optimizing a token budget under budget and format reward signals. **(4) ThinkLess (Fang et al., 2025):** a reinforcement learning-based method that optimizes reasoning by detecting critical thinking points and skipping low-value steps. It introduces a reward function that balances accuracy with token usage, enabling models to "think less" while maintaining performance. We use the same hyperparameters as specified in their respective studies to ensure standardized evaluation conditions, enabling precise comparisons of each efficient reasoning method.

**Results.** As shown in Table 1, FROST achieves the best overall performance across state-of-the-art efficient reasoning methods, delivering slight accuracy improvements while substantially reducing token usage in response generation. Specifically, FROST improves accuracy by an average of 26.70% and reduces token usage by 69.68% on the three base models, GPT-OSS-20B, Magistral-Small-1.1 and Phi-4-reasoning. Although TALE achieves the highest accuracy on certain tasks, this comes at the cost of significantly longer responses. This observation aligns with our assumption that excessively long or overly short responses can degrade model performance. By reducing token usage and focusing on high-attention sentences—i.e., critical reasoning traces—FROST lowers the probability of hallucination or misleading content and grounds responses in essential reasoning. However, FROST may still occasionally prune low-attention but important reasoning steps, which explains why its accuracy is not always the best across all baselines.

## 6.1 Supplementary Experiments

In this section, we conduct additional experiments to examine the influence of our method's performance at different training stages and under different attention functions.

**Efficiency of Different Activation Functions.** To evaluate the contribution of $\text{Softmax}_1$ in FROST, we conduct experiments comparing FROST with different activation functions: vanilla Softmax, Sparsemax (Hu et al., 2023; Martins & Astudillo, 2016), and Entmax15 (Wu et al., 2024; Correia et al., 2019). Here, Entmax15 is a special case of Tsallis $\alpha$-entmax transformations, which interpolate between softmax and sparsemax. We evaluate these strategies on four datasets—GSM8K, MATH500, AIME24, and Minerva—using Phi-4-Reasoning. As shown in Table 2, the results demonstrate that FROST achieves the best overall performance in both Pass@1 accuracy and token usage. Specifically, the average accuracy increases by **15.65%**, while the number of tokens decreases by **68.18%** compared to the base model. FROST also surpasses the overall performance of Sparsemax and Entmax15, which tend to sharpen both low- and high-attention sentences, potentially cutting

Table 2: **Performance of Different Activation Functions.** We evaluate the impact of activation functions on method performance under the same training setup in FROST, using Phi-4-Reasoning across four mathematical datasets (GSM8K, MATH500, AIME24, and Minerva). Pass@1 and token usage (#Tk) are reported as evaluation metrics, with variance omitted since it is consistently $\leq 2\%$. Best results are shown in **bold**, and second-best are underlined. In most settings, FROST achieves the best performance, with $\mathrm{Entmax15}$ consistently ranking second.

| Method | GSM8K | | MATH500 | | AIME24 | | Minerva | | Pass@1 | #Tk |
|---|---|---|---|---|---|---|---|---|---|---|
| | Pass@1 | #Tk | Pass@1 | #Tk | Pass@1 | #Tk | Pass@1 | #Tk | | |
| Base | 0.9242 | 1017.70 | 0.5480 | 1721.95 | 0.0667 | 1017.70 | 0.2500 | 1898.86 | 0.4472 | 1414.05 |
| Softmax | 0.8317 | 1160.63 | 0.4880 | 1379.52 | 0.1333 | 1909.07 | 0.2390 | 1934.72 | 0.4230 | 1595.99 |
| Sparsemax | 0.8188 | 160.99 | 0.5120 | 451.59 | 0.1667 | 948.60 | 0.2647 | 580.84 | 0.4406 | 535.26 |
| Entmax15 | 0.8984 | 163.75 | 0.5520 | 406.97 | 0.1667 | 876.63 | 0.2831 | 439.48 | 0.4751 | 471.71 |
| $\mathrm{Softmax_1}$ (FROST) | 0.9311 | 154.33 | 0.5980 | 344.37 | 0.2667 | 899.80 | 0.2716 | 401.19 | 0.5169 | 449.92 |

off critical reasoning traces. In contrast, FROST is less prone to this issue. The only exception is the Minerva dataset, where $\mathrm{Entmax15}$ attains higher accuracy than FROST while maintaining a similar number of tokens. The underlying reason is difficult to explain at this stage, but it is a pleasant surprise that, except for GSM8K, the overall performance of $\mathrm{Sparsemax}$ and $\mathrm{Entmax15}$ does not decline significantly and in some cases even surpasses the base model. This offers a perspective contrary to that of Yang et al. (2025); Wang (2024).

Table 3: **Outlier Removal Performance in FROST.** We evaluate outlier removal performance on the AIME2024 dataset using the Phi-4-Reasoning model. As outlier metrics, we report the maximum infinity norm $\|\mathbf{x}\|_\infty$ and average kurtosis of the activation tensors. To assess the proportion of critical traces, we also report the average sentence entropy before and after applying FROST. All results are reported with variance omitted, as it is consistently $\leq 2\%$. Best results are shown in **bold**, and second-best results are underlined. In most settings, FROST achieves the best performance in outlier removal and yields higher average sentence entropy. These metrics demonstrate that our method effectively removes reasoning outliers, thereby improving both reasoning performance and efficiency.

| Method | Maximum Infinity Norm $\|\mathbf{x}\|_\infty \downarrow$ | Average Kurtosis $\downarrow$ | Average Sentence Entropy $\uparrow$ | Pass@1 $\uparrow$ | #Tk $\downarrow$ |
|---|---|---|---|---|---|
| Base | 35.31 | 241.72 | 2.71 | 0.0667 | 1017.70 |
| Softmax | 34.53 | 189.36 | 2.79 | 0.1333 | 1909.07 |
| Sparsemax | 34.06 | 152.18 | 2.93 | 0.1667 | 948.60 |
| Entmax15 | 30.39 | 43.72 | 2.92 | 0.1667 | **876.63** |
| FROST | **29.67** | **21.54** | **3.07** | **0.2667** | 899.80 |

**Outlier Removal Performance in FROST.**    To evaluate the performance of FROST in removing attention outliers, we employ two outlier-specific metrics: the *maximum infinity norm* $\|\mathbf{x}\|_\infty$ of the activation tensors $\mathbf{x}$ across all Transformer layers, and the *average kurtosis* of $\mathbf{x}$, which together quantify the presence of outliers. In addition, to demonstrate that removing attention outliers increases the probability assigned to critical sentences, we introduce an entropy-based evaluation metric. Following Wang et al. (2025), token entropy serves as an indicator of criticality: critical tokens tend to exhibit higher entropy than non-critical ones. When a sentence contains more critical tokens, it is expected to exert a stronger influence on final answer generation. Accordingly, we use average sentence entropy to assess whether the reasoning traces in FROST become more critical after training. In our experiments, we analyze these metrics on the AIME2024 dataset using the Phi-4-Reasoning model and compare them with the base model. As shown in Table 3, FROST effectively reduces outliers, evidenced by lower maximum infinity norm $\|\mathbf{x}\|_\infty$ and average kurtosis values. Furthermore, the increase in average sentence entropy indicates that FROST strengthens the model's focus on critical reasoning traces, thereby improving reasoning efficiency. Specifically, we reduce the maximum infinity norm $\|\mathbf{x}\|_\infty$ by **15.97%** and the average kurtosis by **91.09%.** In addition, the average sentence entropy increases by **13.28%** compared to the base model. Additionally, the results show that reasoning outlier metrics—maximum infinity norm $\|\mathbf{x}\|_\infty$ and average kurtosis—are closely related to model performance and average sentence entropy. Higher outlier values correspond to lower sentence entropy and less efficient reasoning traces. This further supports that the reasoning-outlier removal contributes to more efficient reasoning. The only exception is that the average sentence entropy of $\mathrm{Sparsemax}$ is similar to $\mathrm{Entmax15}$, while

the reasoning outlier values of $\mathrm{Entmax15}$ are much smaller than those of $\mathrm{Sparsemax}$. A plausible explanation is that both $\mathrm{Entmax15}$ and $\mathrm{Sparsemax}$ act as sharpening activations that jointly suppress low- and high-valued attention scores. This bidirectional truncation can inadvertently remove parts of crucial reasoning traces, lowering average sentence entropy and reducing Pass@1 performance. Meanwhile, attention outlier metrics such as the maximum infinity norm and kurtosis primarily reflect internal activation dynamics rather than output quality, explaining their relative stability despite external performance declines. Since both activations reshape attention distributions similarly, their outputs also appear alike—with comparable Pass@1 and entropy values—though $\mathrm{Entmax15}$'s smoother contraction yields slightly less degradation in outlier metrics. Overall, this indicates that excessive sharpening can eliminate valuable reasoning signals even while suppressing attention outliers, highlighting $\mathrm{Softmax_1}$'s advantage through selective tail contraction.

## 6.2 GENERALIZABILITY OF MODEL

In this section, we evaluate the generalization ability of FROST on out-of-domain reasoning tasks to verify that its improvements do not harm, but rather preserve or enhance, the model's generation quality beyond the training domain. Using Phi-4-Reasoning as the base model, we test on three additional reasoning benchmarks—LeetCode (Xia et al., 2025b), LiveCodeBench (Jain et al., 2025), and UGPhysical (Xu et al., 2025b)—covering both coding and physical reasoning tasks. The results in Table 4 show that FROST preserves—and even improves—generalization to unseen reasoning tasks. This is expected because FROST filters out uncritical reasoning traces in a manner that generalizes beyond the specific tasks used during fine-tuning. Since FROST only replaces the attention activation with $\mathrm{Softmax_1}$ and uses lightweight LoRA updates, the parameter shift is minimal, ensuring that the model's broader reasoning ability remains intact.

Table 4: **FROST Generalization on Other Reasoning Tasks.** We evaluate the generalization of FROST using Phi-4-Reasoning across three out-of-domain reasoning tasks (Code and Physics). Pass@1 accuracy and token usage (#Tk) are reported, with variance consistently $\leq 2\%$. Best results are in bold, and second-best are underlined. FROST consistently achieves top performance, demonstrating strong generalization across reasoning domains.

| Method | Leetcode | | LiveCodeBench | | UGPhysics | | $\overline{\mathrm{Pass@1}}$ | $\overline{\#\mathrm{Tk}}$ |
|--------|----------|------|---------------|------|-----------|------|--------|------|
| | Pass@1 | #Tk | Pass@1 | #Tk | Pass@1 | #Tk | | |
| Base | 0.3222 | 2755.13 | 0.3248 | 3154.80 | 0.3172 | 2603.00 | 0.3214 | 2837.64 |
| Softmax | 0.3778 | 2106.85 | 0.3538 | 2909.07 | 0.3011 | 2622.52 | 0.3442 | 2546.15 |
| FROST | **0.3889** | **1163.06** | **0.3777** | **1967.56** | **0.3473** | **805.77** | **0.3713** | **1312.13** |

## 7 DISCUSSION AND CONCLUSION

We propose an attention-aware efficient reasoning method, **FROST**. Our approach introduces the concept of *reasoning outliers*, which contribute to high latency in reasoning performance, and provides an outlier removal mechanism that enables LRMs to leverage lightweight supervised fine-tuning (SFT) for generating efficient and accurate reasoning traces in mathematical problem solving. Theoretically, we show that the existing outlier removal technique $\mathrm{Softmax_1}$ is effective for reasoning outliers and can operate at the sentence level. Empirically, FROST improves response accuracy by **26.70%** and reduces token usage by **69.68%**. In addition, it decreases the maximum infinity norm $\|\mathbf{x}\|_\infty$ by **15.97%** and average kurtosis by **91.09%**, confirming its effectiveness in mitigating reasoning outliers.

Although FROST achieves strong performance in efficient reasoning, several limitations remain. First, our method is currently restricted to mathematical reasoning tasks, while many reasoning models also target domains such as coding. Second, FROST relies solely on supervised fine-tuning and does not incorporate GRPO, which could further enhance efficiency. In future work, we plan to extend FROST to additional reasoning tasks, including coding, and to develop a GRPO-based approach that builds on our current findings to further improve efficient reasoning performance.

## ACKNOWLEDGMENTS

The authors would like to thank Dennis Wu for insightful discussions on theoretical analysis. The authors would like to thank the anonymous reviewers and program chairs for constructive comments. Haozheng Luo is partially supported by the Lambda Research Grant. This research was supported in part by the computational resources and staff contributions of the RTRC High Performance Computing Cluster. Additional support was provided by the Quest High Performance Computing Facility at Northwestern University, which is jointly supported by the Office of the Provost, the Office for Research, and Northwestern University Information Technology. The content is solely the responsibility of the authors and does not necessarily represent the official views of the funding agencies.

## ETHICAL STATEMENT

This work investigates reasoning outliers in large reasoning models (LRMs) and proposes an outlier-removal technique to mitigate attention outliers. In line with the ICLR Code of Ethics[*] , we acknowledge that our method may inadvertently amplify biases present in training data, potentially leading to unfair outcomes for underrepresented groups. Prior studies have also noted that supervised fine-tuning (SFT) can induce shallow alignment and affect red-teaming protection, but this issue is outside the scope of our work. Our focus is on improving reasoning efficiency, and we believe this research does not raise serious ethical concerns.

## REPRODUCIBILITY

All experiments are conducted with three random seeds, yielding stable results with standard deviations below 2%. We adopt a unified training setup using the AdamW optimizer with learning rate $1 \times 10^{-5}$, batch size 8, and batch size 256 for deployment. For low-rank adaptation, we set the LoRA rank to 8 and LoRA $\alpha$ to 16. Detailed hyperparameters are provided in Appendix E.2, and theoretical proofs are included in Appendix C.

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

# Supplementary Material

## A  IMPACT STATEMENT

We believe this methodology offers an opportunity to strengthen the core of large reasoning models by improving efficiency and enabling models to produce more critical reasoning traces. However, it may also amplify biases present in the training data, potentially leading to unfair or discriminatory outcomes for underrepresented groups.

## B  ADDITIONAL RELATED WORK

**Efficient Alignment.** In recent years, foundation models (Zhou et al., 2025; He et al., 2025; 2024; Wang et al., 2024; Touvron et al., 2023) have shown strong capabilities in solving multitask problems. To further improve their performance on specific tasks, alignment techniques are essential for refining model behavior. However, traditional approaches like RLHF (Ouyang et al., 2022) and DPO (Rafailov et al., 2023) are computationally expensive. This highlights the urgent need for parameter-efficient fine-tuning methods that offer effective and economical alignment for foundation models. Several traditional methods demonstrate strong capabilities in aligning foundation models, including LoRA (Hu et al., 2021) and QLoRA (Dettmers et al., 2023). Building on this, Luo et al. (2025b) propose a LoRA variant that replaces the standard softmax layer with OutEffHop layers (Hu et al., 2024) to improve the efficiency of low-rank adaptation. However, all of these methods are heavily based on LoRA, and when adaptation is required for modules outside the attention architecture, the computational cost increases significantly. Zhao et al. (2025b); Luo et al. (2025c) propose novel alignment methods that focus on small subsets of neurons within foundation models. For example, Zhao et al. (2025b) identify key neurons with high influence on LLMs' jailbreak defense using latent representations, and fine-tune only these neurons using red-teaming datasets. Our method builds on fast low-rank adaptation techniques (Luo et al., 2025b), further improving adaptation efficiency, and integrates them into SFT training to optimize reasoning paths and produce efficient reasoning models.

## C    PROOFS OF MAIN TEXT

### C.1    LEMMA 5.1

*Proof of Lemma 5.1.* Monotonicity means that if we increase any input coordinate to $\phi$, its output does not decrease. Let $u = \{z_t\}_{t \in S_i}$ and $w = \{z_{t'}\}_{t' \in S_j}$. If for each coordinate of $u$ there is a not-smaller coordinate in $w$ replaced by the smaller value, then by repeatedly applying coordinatewise monotonicity we obtain $s_i = \phi(u) \geq \phi(w) = s_j$. Order preservation (P1) then yields $\alpha_i \geq \alpha_j$.

### C.2    THEOREM 5.1

*Proof of Theorem 5.1.* By Lemma 5.1, sentence scores $s$ reflect dominance induced by token compatibilities under $\phi$. Applying Assumption 5.1(P3) directly to $s$ yields (5.1). Assumption 5.1(P2) allows re-centering $s \leftarrow s - c\mathbf{1}$ without changing $\alpha$; thus (5.1) is invariant to any global shift and depends only on relative separations.

### C.3    THEOREM 5.2

*Proof of Theorem 5.2.* For (5.2), apply operator-norm submultiplicativity: $\|W_o(\alpha_i v_i)\| \leq \|W_o\|_{\mathrm{op}} \cdot \alpha_i \|v_i\| \leq B_o \,\varepsilon\, B_v$. To obtain (5.3), propagate the perturbation through $L$ differentiable layers with Jacobians $J_\ell$:

$$\|\Delta \ell_i^{(L)}\| \; \leq \; \Big(\prod_{\ell=1}^{L} \|J_\ell\|_{\mathrm{op}}\Big) \|W_o\|_{\mathrm{op}} \, \alpha_i \|v_i\| \; \leq \; \varepsilon\Big(\prod_{\ell=1}^{L} B_\ell\Big) B_o B_v.$$

Finally, since $\mathrm{Softmax}_1$ is 1-Lipschitz in $\ell_\infty \to \ell_1$, the change in probabilities is bounded by the logit change, yielding (5.4). Replacing $\prod_{\ell=1}^{L} B_\ell$ with $B^L$ (by definition of $B$) gives the stated $O(B_o B_v B^L \varepsilon)$ rate. If $B_o, B_v, B$ are $O(1)$, the rate simplifies to $O(\varepsilon)$.

## D    AN EXAMPLE OF LRM REASONING TRACES

In this section, we analyze the Phi-4-Reasoning response to the first question of AIME24, which is also illustrated in Figure 4. As shown in the color box in Appendix D, traces S1 and S2 are classified as uncritical. Although S2 includes partially critical content such as `"So the walking time (actual walking time) plus t minutes equals total time."`, its overall reasoning remains non-critical. Trace S3 represents a critical reasoning step, where the model identifies the two key equations in the problem. Subsequently, from S4 to S19, the model enters a self-verification phase, producing reasoning traces beginning with `wait` that reflect self-checking and correction. Starting from S20, the model resumes critical reasoning after the signal `"We'll produce final answer in a box."`, and by S24, it generates the final answer, concluding its reasoning process.

## E    EXPERIMENT SYSTEM AND IMPLEMENT SETTINGS

### E.1    COMPUTATIONAL RESOURCES

We perform all experiments using two NVIDIA H100 GPUs with 80GB of memory and a 12-core INTEL(R) XEON(R) PLATINUM 8592 CPU operating at 1.90GHz. Our code is developed in PyTorch and utilizes the Hugging Face Transformer Library for experimental execution. For running the LLMs, we use the default system prompt provided by the official source and set the temperature to 0.6 to balance consistency and performance.

### E.2    HYPERPARAMETERS

We present the hyperparameters used in the fine-tuning stage for each model. We use **AdamW** (Loshchilov & Hutter, 2019) as the optimizer. Most other hyperparameters are kept consistent across all models and datasets, including a batch size of 256 during deployment and 8 during training. In training, we also use gradient accumulation with 4 steps and set the weight decay to 0.01 for

---

**An Example of LRM Reasoning Traces**

**[Question]** Every morning Aya walks $9$ km and then spends $t$ minutes in a coffee shop. At speed $s$ km/h, the total time is $4$ hours (including the coffee stop). At speed $s + 2$ km/h, the total time is 2 hours 24 minutes. If she walks at $s + \frac{1}{2}$ km/h, find the total time in minutes (including the coffee stop).

**[Uncritical Trace S1]** <think>We are given: "Every morning Aya goes for a 9-kilometer-long walk and stops at a coffee shop afterwards. When she walks at a constant speed of $s$ kilometers per hour, the walk takes her 4 hours, including $t$ minutes spent in the coffee shop. When she walks $s + 2$ kilometers per hour, the walk takes her 2 hours and 24 minutes, including $t$ minutes spent in the coffee shop. Suppose Aya walks at $s + \frac{1}{2}$ kilometers per hour. Find the number of minutes the walk takes her, including the $t$ minutes spent in the coffee shop."

**[Uncritical Trace S2]** Let's denote: "Walking distance = 9 km. There's a coffee shop stop time $t$ minutes, but note that the given total times includes the coffee shop stop time. So the walking time (actual walking time) plus $t$ minutes equals total time. Let $s$ be her walking speed (km/h) in first scenario. Then walking time (in hours) is $9/s$. And the total time is 4 hours."

**[Critical Trace S3]** But includes $t$ minutes stop, so actual walking time is 4 hours minus $t$ minutes (converted to hours, $t/60$ hours), so equation: $9/s = 4 - t/60$. (Equation 1) and $9/(s + 2) = 2.4 - t/60$ (Equation 2)

**[Uncritical Trace S4]** But wait, let's check: speed $= s + 2$, then walking time (in hours) is $9/(s + 2)$. Total time is 2 hours 24 minutes, which is $2 + 24/60 = 2.4$ hours.

**[Uncritical Trace S5]** But wait, let's check: The problem said: "Suppose Aya walks at s+1/2 kilometers per hour. Find the number of minutes the walk takes her, including the t minutes spent in the coffee shop."

**[Uncritical Trace S6]** Wait, 2 hours 24 minutes = 144 minutes.

**[Uncritical Trace S7]** Let me recalc: 3 hours 24 minutes = 2*60 + 24 = 120 + 24 = 144 minutes.

**[Uncritical Trace S8]** Let us double-check the reasoning process.

**[Uncritical Trace S9-S17]** .....

**[Uncritical Trace S18]** I'll now produce step-by-step explanation.

**[Uncritical Trace S19]** We'll produce final answer in a box.

**[Critical Trace S20]** The two equations are given as $9/s = 4 - t/60$ (Equation 1) and $9/(s + 2) = 2.4 - t/60$ (Equation 2).

**[Critical Trace S21]** Alternatively, we can solve for $t$ in terms of $s$ from equation 1: $t = 60 * (4 - 9/s) = 240 - 540/s$, and plug in equation 2: $9/(s+2) = 2.4 - (240 - 540/s)/60 = 2.4 - (4 - 9/s) = 2.4 - 4 + 9/s = (9/s) - 1.6$.

**[Critical Trace S22]** So equation becomes: $9/(s + 2) = (9/s) - 1.6$. Multiply both sides by $s(s + 2)$: $9s = (9(s + 2)) - 1.6s(s + 2)$.

**[Critical Trace S23]** Now, when Aya walks at speed $s + 1/2 = 2.5 + 0.5 = 3$ km/h, then walking time (in hours) is $9/3 = 3$ hours. But total time includes coffee shop stop time $t$ minutes, which is 24 minutes $= 24/60 = 0.4$ hours. So total time is $3 + 0.4 = 3.4$ hours, which is 3 hours 24 minutes.

**[Critical Trace S24]** I'll produce final answer: $\boxed{204}$. </think>

---

all training runs. A learning rate of $1e^{-5}$ is used for all models during fine-tuning. For low-rank adaptation, we use a LoRA rank of 8 and LoRA alpha set to 16. In FROST, we set the maximum training steps to 5,000. All supervised fine-tuning and GRPO training are conducted using mixed precision with bfloat16. In deployment, we set the temperature to 0.6 for all models with top-$p$ sampling at 0.9. For evaluation, we use a maximum generation length of 4096 across all models, except TALE.

## F ADDITIONAL EXPERIMENTS

In this section, we present additional experiments demonstrating that FROST surpasses current state-of-the-art efficient reasoning methods.

### F.1 TRAINING AND TEST TIME COMPARISON

We conduct experiments to measure the training and inference time of each baseline and compare their computational costs with FROST. For evaluation, test time is measured on the AIME tasks with the GPT-OSS-20B model, while training time is reported using the respective datasets specified in each baseline's original paper. All experiments are conducted on the same computational resources, as described in Appendix E.1.

Table 5: **Comparison of Training and Test Time Costs Across Methods.** We conduct experiments to measure the training and test time of each method. For test-time evaluation, we use the AIME dataset with the GPT-OSS-20B model. Best results are shown in **bold**, and second-best are underlined.

| Method | TALE | DRP | ThinkLess | FROST |
|---|---|---|---|---|
| Training Time (m) | - | 353 | 1186 | **204** |
| Test Time (m) | 56 | 18.5 | 4.2 | **3** |

As shown in Table 5, FROST achieves the fastest training time among all methods, while also minimizing computation cost and inference time during deployment. This demonstrates that our approach not only accelerates training but also reduces deployment overhead.

### F.2 ATTENTION DISTRIBUTIONS OF ACTIVATION FUNCTIONS

We conduct an additional experiment to analyze the attention distribution of GPT-OSS-20B on a sample from the GSM8K dataset. As shown in Figure 6, FROST effectively removes a large number of low-attention sentences while retaining significant ones. In contrast, the vanilla model produces many sentences with low attention weights, and $\mathrm{Sparsemax}$ and $\mathrm{Entmax15}$ retain only one to two sentences, often aggressively discarding important reasoning traces. This visualization provides an explanation consistent with the performance results reported in Table 2.

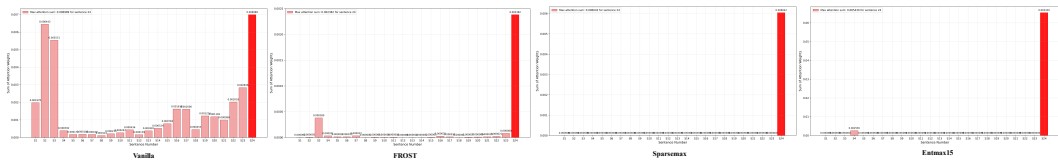

Figure 6: **Attention Distribution of Each Activation Function.**

## G INFLUENCE THE ATTENTION DYNAMICS OF $\mathrm{Softmax}_1$ DURING TRAINING AND INFERENCE

We observe that incorporating $\mathrm{Softmax}_1$ significantly influences both training and inference attention dynamics across transformer layers. During supervised fine-tuning (SFT), $\mathrm{Softmax}_1$ enforces tail contraction by suppressing low-attention activations, which stabilizes gradients and reduces the variance of updates propagated through residual connections. This effect leads to faster convergence of LoRA adapters, as the low-rank parameter subspace more efficiently aligns with critical attention directions, improving overall adaptation coverage within fewer training steps. This observation is consistent with Luo et al. (2025b); Hu et al. (2024). Across layers, $\mathrm{Softmax}_1$ reshapes the attention landscape—shallow layers become more selective in contextual grounding, while deeper layers exhibit higher entropy concentration around critical reasoning traces. During inference, this sharpening propagates forward, effectively filtering redundant reasoning sentences while maintaining coherence. Together, these behaviors demonstrate that $\mathrm{Softmax}_1$ not only enhances efficient reasoning but also accelerates LoRA-SFT optimization by improving the representational focus of each attention head.

## H  Influence of $\text{Softmax}_1$ Across Layers

We analyze the effect of $\text{Softmax}_1$ across transformer layers by visualizing the attention distributions of head 15 for both vanilla $\text{Softmax}$ and $\text{Softmax}_1$. As shown in Figures 7 and 8, $\text{Softmax}_1$ consistently suppresses attention outliers, leading to smoother and more stable activations across the network. In lower layers, $\text{Softmax}_1$ contracts heavy tails and mitigates rare extreme peaks, enhancing local feature mixing with higher-entropy and reduced kurtosis distributions. In higher layers, it suppresses residual long-range spikes and sharpens focus on semantically relevant tokens, yielding sparser yet more stable attention and clearer causal information flow.

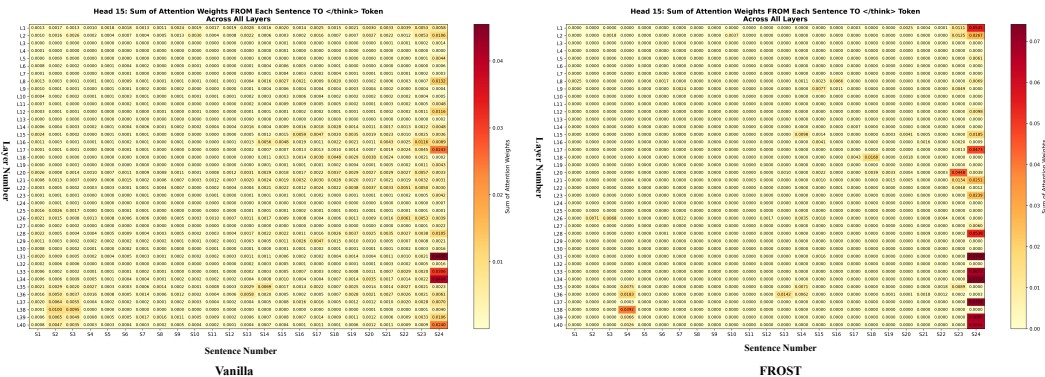

Figure 7: **Theoretical Analysis of Reasoning Outlier Removal in All Layers**

## I  Extended Attention Heatmaps Across Additional Layers and Heads

In this section, we present extended attention heatmaps covering additional layers and heads. Specifically, we analyze **Layers 0, 5, 15, 25, 30, 35, and 39** and **Heads 0, 5, 10, 15, 20, 25, 30, 35, and 39** to provide a more comprehensive view of attention evolution across the network. The corresponding observations are illustrated in Figure 9.

## J  Human Expert Evaluation

We invite three computer science students specializing in reasoning models to annotate reasoning traces generated by the original and FROST-trained models. We then compare the traces pruned by FROST and evaluate their criticality based on relevance and contribution to the final answer. Averaging across all evaluators, FROST achieves **92%** accuracy in correctly removing non-critical reasoning traces. Only **8%** of reasoning traces are incorrectly removed, which significantly degrades final-answer accuracy. These mistakenly pruned traces are typically long and contain repeated information that supports self-verification and error correction. However, they also provide critical content—such as key equations—in the end of trace. This observation suggests a potential explanation for why FROST achieves the second-best Pass@1 score in the Phi-4-Reasoning experiment shown in Table 1.

## K  Disclosure of LLM Usage

In our paper and project, we use large language models (LLMs) to help revise the text for greater conciseness and precision.

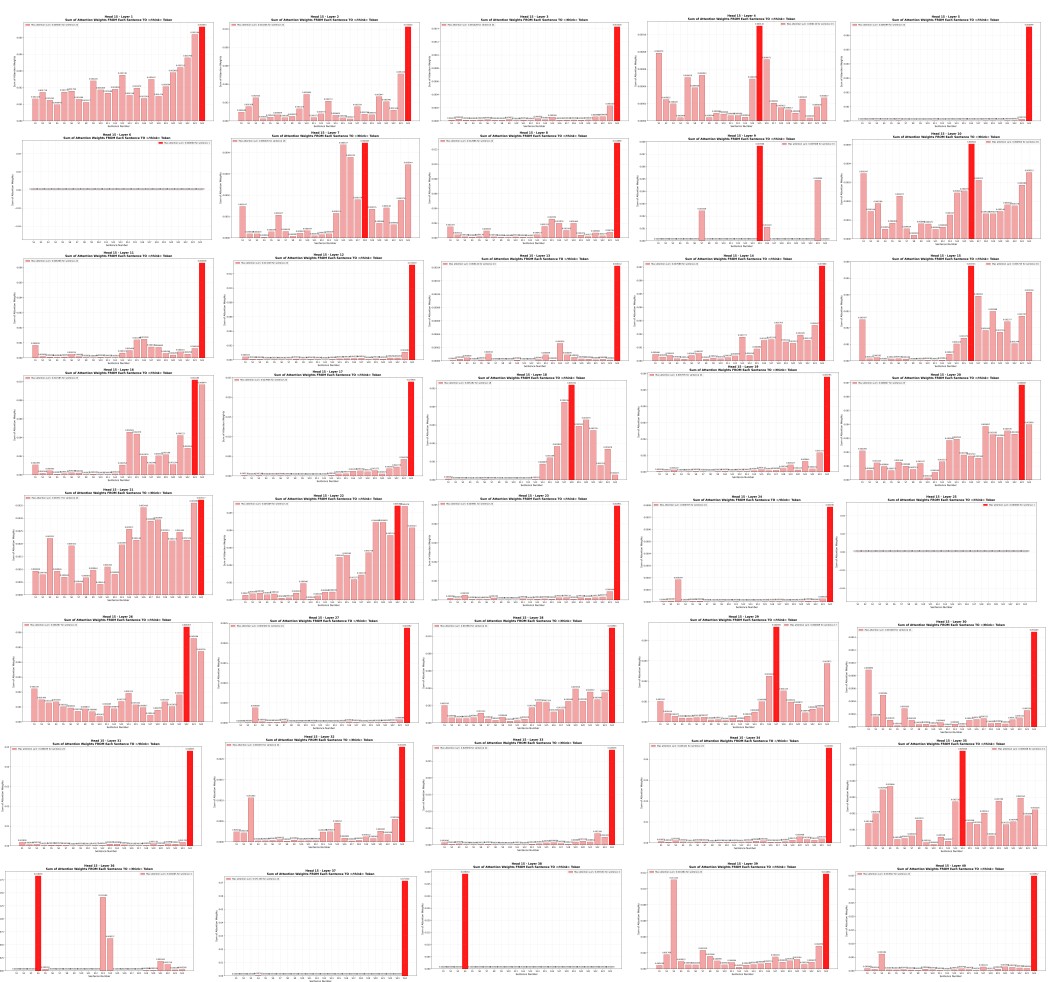

Figure 8: **Attention Distribution of** $\mathrm{Softmax}_1$ **Across All Layers**

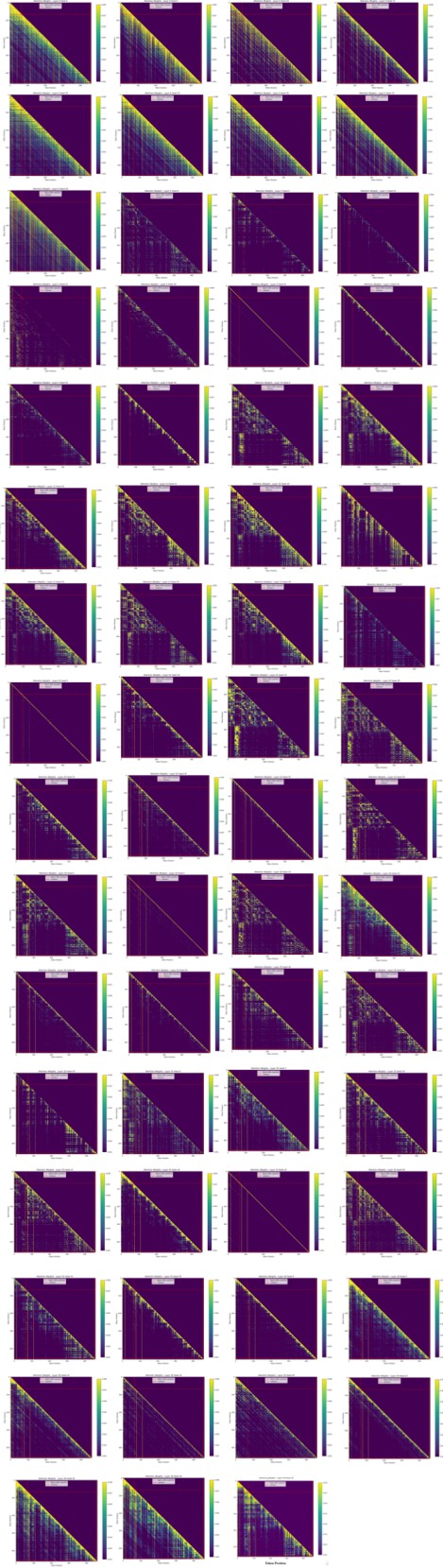

Figure 9: **Extended Attention Heatmaps Across Additional Layers and Heads**

