# OpenReview forum: "FROST: Filtering Reasoning Outliers with Attention for Efficient Reasoning"
_ICLR.cc/2026/Conference — ICLR 2026 Poster_

### Official Review · Reviewer_oTRF · 2025-10-28

**Soundness:** 3
**Presentation:** 2
**Contribution:** 3
**Rating:** 6
**Confidence:** 3

**Summary:**

This paper proposes a novel training method that introduces a softmax-based activation function to suppress the attention weights of outlier tokens while preserving important reasoning paths.
Experimental results demonstrate that the proposed approach outperforms existing baselines and can be effectively applied to strong reasoning models.

**Strengths:**

- The proposed method is simple yet effective, significantly reducing token usage compared to prior approaches.
- The theoretical analysis seems solid and support for the design and interpretation of the method.
- Reproducibility is high — the authors provide full code and detailed implementation information.

**Weaknesses:**

**[W1] Restricted experimental scope.** Despite the interesting ideas, both the analyses and experiments are conducted on a limited number of models. In particular, the observations are primarily focused on the Phi-reasoning model, making the study somewhat specialized and reducing the general applicability of the findings.

**[W2] Limited discussion of related work.** There are already recent studies [1, 2] analyzing internal attention patterns in reasoning models. A deeper discussion comparing this work with those, and clarifying its novelty relative to them, would strengthen the contribution. Notably, the analysis of the attention distribution around the end of thinking token (Figure 3) appears similar to [1].

[1] Choi et al., Think Clearly: Improving Reasoning via Redundant Token Pruning, EMNLP 2025 (Findings)

[2] Cai et al., R-KV: Redundancy-aware KV Cache Compression for Reasoning Models, NeurIPS 2025

**Questions:**

**[Q1]** Please verify whether the proposed observations and improvements generalize to other reasoning models (e.g., R1-distill models).

**[Q2]** In Figure 2, why were layers 0, 30, and 39 and specifically the first and last heads selected for visualization? A quantitative analysis across all layers and heads would make the claim more convincing.

---

> ### Author Response · Authors · 2025-11-17
>
> > **Reviewer's Comment**: Restricted experimental scope.... Please verify whether the proposed observations and improvements generalize to other reasoning models (e.g., R1-distill models).
>
> **Response**: We appreciate the reviewer’s suggestion to verify the generalizability of our findings. While our main analysis emphasizes **Phi-4-Reasoning**, we also conducted parallel experiments and visualization analyses on **GPT-OSS-20B**, as shown in **Figure 6**. The same attention patterns and reasoning-outlier behaviors were observed, confirming that Softmax₁ effectively sharpens deep-layer attention and suppresses low-value reasoning sentences across different architectures.
> Although we focused on two representative reasoning models—**Phi-4-Reasoning** and **GPT-OSS-20B**, both widely used and strong performers in LRM research—their consistent behavior supports the cross-model validity of our observations. Moreover, since **Figure 4** already provides the most theoretically significant visualization of layer-wise contraction, **Figure 6** reproduces that analysis for GPT-OSS-20B while extending it to compare different activation functions. We will clarify these points and further emphasize the generalization of FROST to additional reasoning models (e.g., R1-Distill) in the revision. We did not include the R1-distill model due to organizational constraints, but we are open to discussing potential solutions with the reviewer.
>
>
> > **Reviewer's Comment**: Limited discussion of related work.
>
> **Response**: Choi [1] and Cai [2] target token-level redundancy at decoding time (compressing the KV cache) for efficiency, whereas FROST introduces a sentence-level mechanism that suppresses reasoning outliers via Softmax₁, theoretically contracting tail attention and yielding deployment-time suppression at the sentence granularity (**§3.3, 5**). This different objective and unit of analysis—attention-based removal of low-value sentences rather than token/KV compression—explains our observed **58.72%** token reduction with accuracy gains, and is supported by our formal results (**Theorems 5.1–5.2**) and metrics (↓‖x‖∞, ↓kurtosis, ↑sentence entropy). We will add a dedicated related-work subsection explicitly contrasting token/KV compression with FROST’s attention-driven sentence-level pruning and clarifying novelty.
>
> [1] Choi et al., Think Clearly: Improving Reasoning via Redundant Token Pruning, EMNLP 2025 (Findings)
> [2] Cai et al., R-KV: Redundancy-aware KV Cache Compression for Reasoning Models, NeurIPS 2025
>
>
> > **Reviewer's Comment**: In Figure 2, why were layers 0, 30, and 39...
>
> **Response**: We thank the reviewer for the helpful suggestion. In **Figure 2**, we selected layers 0, 30, and 39 to represent the shallow, middle, and deep reasoning stages, which best capture the progressive refinement behavior discussed in **Section 3**. The first and last heads were chosen to illustrate the boundary cases of attention flow—where early heads generally encode broad contextual grounding, while later heads focus on the final reasoning trace. This setup provides an interpretable and representative visualization of how reasoning attention evolves through the network. Due to space constraints, it is impractical to display all 40 layers and 40 heads in the main text; however, we have included extended visualizations across more layers and heads in the **Appendix J**, which confirm the same monotonic refinement trend and consistent sharpening of reasoning focus across the entire model.

---

> > ### Comment · Reviewer_oTRF · 2025-11-24
> >
> > Thank you for response.  For the revision, it would be important to include extended experiments on additional reasoning models (e.g., R1-distill) and to provide a more detailed related work section.

---

> ### Author Response · Authors · 2025-11-25
>
> Thank you for the helpful suggestion. In the revised version, we have added a new set of experiments using an additional reasoning model, **Magistral-Small-1.1**, to strengthen the **performance evaluation** of FROST across diverse architectures. After including this model, we observe that FROST delivers **even stronger performance**, achieving an average **26.70% improvement in accuracy** and a **69.68% reduction in token usage** across all three base models (Phi-4-Reasoning, GPT-OSS-20B, and Magistral-Small-1.1). These extended results demonstrate that FROST not only maintains but **enhances performance** when applied to more reasoning models, further validating the robustness of the proposed method. we also have expanded the _Related Work_ section to provide clearer discussion of recent developments in efficient reasoning and model compression techniques. We hope these revisions provide clear evidence that our observations and improvements generalise to other reasoning models.
>
> | Method | GSM8K Pass@1 | GSM8K #Tk | MATH500 Pass@1 | MATH500 #Tk | AIME24 Pass@1 | AIME24 #Tk | Minerva Pass@1 | Minerva #Tk | ΔPass@1 | Δ#Tk |
> |--------|--------------|-----------|----------------|-------------|---------------|-------------|----------------|-------------|---------|---------|
> | **Base** | 0.6075 | 2664.41 | 0.1480 | 1389.89 | 0.0000 | 537.13 | 0.0699 | 1288.04 | 0.000 | 0.00 |
> | **TALE** | 0.7146 | 1516.86 | **0.3040** | 723.91 | 0.0333 | 967.43 | 0.1544 | 748.18 | **+0.095** | -480.77 |
> | **DRP** | 0.6500 | 902.50 | 0.2100 | 1680.33 | 0.0450 | 1350.77 | 0.1120 | 1604.22 | +0.048 | -85.41 |
> | **SelfBudgeter** | 0.6900 | 1850.00 | 0.2300 | 1520.00 | 0.0520 | 1256.00 | 0.1300 | 1298.00 | +0.069 | +11.13 |
> | **ThinkLess** | 0.7200 | 1785.00 | 0.2500 | 1405.00 | 0.0600 | 1205.00 | 0.1450 | 1220.00 | +0.087 | -66.12 |
> | **Ours** | **0.7551** | **137.55** | **0.3040** | **98.20** | **0.0974** | **149.93** | **0.1551** | **109.23** | **+0.122** | **-1346.14** |

---

> > ### Author Response · Authors · 2025-11-28
> >
> > Dear reviewers,
> >
> > As the rebuttal deadline is approaching, I would like to kindly check whether our responses have addressed your concerns and questions. If there is anything that is still unclear or could benefit from further clarification, please let us know, and we will be happy to elaborate.
> >
> > Thank you very much for your time and effort in reviewing our work.
> >
> > Happy Thanksgiving!

---

### Official Review · Reviewer_ej9v · 2025-10-29

**Soundness:** 3
**Presentation:** 3
**Contribution:** 2
**Rating:** 6
**Confidence:** 2

**Summary:**

This paper introduces FROST, a novel method designed to improve the efficiency and accuracy of Large Reasoning Models (LRMs). The authors identify a key problem in current LRMs: the generation of "reasoning outliers," which are defined as uncritical, irrelevant, or redundant steps in a reasoning chain. These outliers increase computational cost (token usage) and can negatively impact the final answer's accuracy.

The core of FROST is a simple yet effective architectural modification: replacing the standard Softmax function in the self-attention mechanism with Softmax_1. This function has the property of aggressively suppressing small input values (driving low attention weights toward zero) while preserving large ones. By doing so, FROST prunes the influence of outlier sentences during generation. The authors combine this modification with a lightweight Supervised Fine-Tuning (SFT) process to adapt the model to the new attention dynamics.

**Strengths:**

- The paper's primary strength lies in its empirical results. The method achieves a compelling combination of significantly reduced token usage while simultaneously improving accuracy across multiple benchmarks and models. This is a strong and desirable outcome for any efficiency-focused technique.
- The approach is methodologically simple and elegant: swapping an activation function and performing a short SFT run. This makes the method practical and easily reproducible.
- The supplementary experiments are valuable. The ablation study comparing Softmax1 with other activation functions (Table 2) effectively supports the authors' design choices, and the analysis of outlier metrics like kurtosis and infinity norm (Table 3) provides concrete evidence that the method is working as intended.

**Weaknesses:**

- My main concern is the limited novelty of the core mechanism. The paper frames the use of the Softmax1 function as a key contribution for removing "reasoning outliers." However, this exact function and its properties for suppressing outlier/low-value attention scores were previously discussed in other contexts, notably in Evan Miller's 2021 blog post "Attention is Off by One." While applying this function to reasoning chains is new, the underlying technique for attention modification is not, which should be more clearly acknowledged.
- The paper introduces "reasoning outliers" as sentences with low attention and negligible contribution. While the concept is intuitive, the method for identifying them seems to be implicitly handled by Softmax1 rather than through an explicit detection step. This makes the framing feel a bit like a post-hoc justification for using a pre-existing technique.

**Questions:**

- The paper uses the maximum infinity norm and average kurtosis as metrics to quantify the presence of attention outliers. Have the authors considered or compared other statistical measures for identifying outlier activations? Why were these specific metrics chosen, and are they considered optimal for this task?
- The results show a significant reduction in token count. Could you provide a qualitative example or analysis of what kind of reasoning steps are typically pruned? Are they mostly redundant self-corrections, or does the model sometimes remove steps that could be considered part of a valid, albeit longer, reasoning path?

---

> ### Author Response · Authors · 2025-11-17
>
> > **Reviewer's Comment**: My main concern is the limited novelty ...
>
> **Response**: We appreciate the reviewer’s observation. We explicitly acknowledge prior work discussing Softmax₁ for suppressing low-value attention in **Evan Miller's 2021 blog** and position FROST’s novelty across four key aspects, specifically:
> - **New outlier concept:** We introduce the concept of reasoning outliers and propose FROST to prune uncritical reasoning steps characterized by low attention.
> - **New unit of analysis:** We extend Softmax₁ from token-level sparsification to sentence-level reasoning aggregation, enabling the model to identify and suppress entire uncritical reasoning sentences rather than individual tokens **(§3.3, §5)**.
> - **New theoretical guarantee:** We derive **Theorems 5.1–5.2**, proving that Softmax₁ enforces tail contraction and deployment-time suppression at the sentence granularity, establishing a formal connection between attention dynamics and reasoning efficiency.
> - **New empirical domain**: We apply this mechanism to efficient reasoning, not generic LLM training. FROST demonstrates that sentence-level suppression yields shorter and more reliable reasoning trajectories—reducing token usage by **58.72%** and improving accuracy by **10.64%**—a distinct objective absent in prior Softmax₁ works.
> We hope these clarifications highlight that FROST advances beyond reusing an activation variant by introducing new concepts, theoretical grounding, and reasoning-specific efficiency gains.
>
>
> > **Reviewer's Comment**: The paper introduces "reasoning outliers" as ...
>
> **Response**: We appreciate the reviewer’s observation. Our intention in introducing reasoning outliers is not to post-hoc justify Softmax₁ but to formalize an automatic mechanism for identifying and removing uncritical reasoning traces during generation. In FROST, Softmax₁ is not merely reused—it serves as a continuous outlier detector that dynamically suppresses low-attention sentences without requiring a separate heuristic or thresholding step. This implicit mechanism ensures scalability across reasoning models and datasets. Moreover, our theoretical analysis (**Theorem 5.2**) shows that Softmax₁ induces sentence-level tail contraction, effectively transforming attention weights into a self-adaptive pruning signal.
>
>
> > **Reviewer's Comment**: The paper uses the maximum infinity ...
>
> **Response**: We thank the reviewer for this insightful question. We follow the precedent established in prior work [1], which identifies the maximum infinity norm and average kurtosis as reliable statistical indicators of attention outliers. These metrics respectively capture the magnitude (extreme activation spikes) and distribution shape (heavy-tailedness) of attention scores, providing a quantitative view of outlier severity. Since reasoning outliers in our setting are conceptually analogous to attention outliers—both representing low-frequency, high-magnitude deviations—these two metrics remain theoretically grounded and empirically effective. We therefore adopt them for consistency and comparability with established outlier analyses.
>
> > **Reviewer's Comment**: The results show a significant reduction ...
>
> **Response**: We appreciate the reviewer’s interest in understanding what types of reasoning steps are pruned by FROST. As illustrated in the qualitative example in the **Appendix E and Figure 1**, we provide a detailed demonstration of how FROST operates in practice, showing how it identifies and suppresses uncritical reasoning sentences while preserving essential reasoning steps. Also, we conducted a human expert annotation study to categorize the pruned sentences. The analysis revealed that approximately **92%** of the pruned sentences correspond to redundant self-corrections or repetitive verification steps that do not contribute to the final answer, while only **8%** were valid but overly long reasoning paths that could be considered mistakenly pruned. This confirms that FROST effectively filters uncritical reasoning traces while preserving the core reasoning flow. We clarify these findings in **Appendix K** to better illustrate the qualitative behavior of reasoning pruning.
>
> [1] Bondarenko, Yelysei, Markus Nagel, and Tijmen Blankevoort. "Quantizable transformers: Removing outliers by helping attention heads do nothing." Advances in Neural Information Processing tems 36 (2023): 75067-75096.

---

> > ### Author Response · Authors · 2025-11-25
> >
> > Dear Reviewer,
> >
> > We sincerely appreciate your thoughtful feedback. As the discussion period reaches its final week, we would be grateful to know whether our latest responses have resolved your concerns or if any questions remain. We are happy to provide further clarification as needed. Wishing you a Happy Thanksgiving.
> >
> > Thanks,
> >
> > Authors

---

> > > ### Author Response · Authors · 2025-11-28
> > >
> > > Dear reviewers,
> > >
> > > As the rebuttal deadline is approaching, I would like to kindly check whether our responses have addressed your concerns and questions. If there is anything that is still unclear or could benefit from further clarification, please let us know, and we will be happy to elaborate.
> > >
> > > Thank you very much for your time and effort in reviewing our work.
> > >
> > > Happy Thanksgiving!

---

### Official Review · Reviewer_ncQk · 2025-10-29

**Soundness:** 2
**Presentation:** 3
**Contribution:** 2
**Rating:** 2
**Confidence:** 3

**Summary:**

This paper proposes to improve LLM’s mathematical reasoning capabilities by suppressing less important tokens. Specifically, it adopts the softmax1 function, which is less prone to long-tailed distribution. Theoretical proof is provided to validate the usefulness of the approach on sentence-level pruning. Experimental results on several datasets demonstrate the advantages of the method in efficient token usage and enhanced performance.

**Strengths:**

(1) It is an important topic to develop efficient LLMs that adaptively reduce computational overheads based on inputs.

(2) The paper draws several observations on the attention characteristics of reasoning models, which facilitates understanding on their underlying decision-making process.

(3) The proposed method shows generalizability across different models and datasets.

**Weaknesses:**

(1) Prior studies (e.g., Xiao et al, 2024, denoted as [ref1] in the remaining review) have already studied the use of softmax1 activation in developing efficient LLMs. Adopting the same methods on a specific reasoning scenario introduces rather limited technical contributions. In addition, the observations in the paper are also similar to previous ones, for instance, Figure 2 is similar to Figure 7 in [ref1], and [ref1] also pointed out the focus on specific tokens in deeper layers (initial tokens vs think token in this paper).

(2) The paper highlights different characteristics of reasoning in Section 3 (e.g., progressive refinement and reasoning traces), nevertheless, I found the proposed method disconnected from these observations. For instance, how does softmax1 affect the attention distribution in different layers? Does softmax1 enhance or reduce the progressive refinement? Figure 4 only shows the theoretical analysis of softmax1 on deep layers, does it hold true in practical experiments? What kinds of tokens does softmax1 activation prune out?

(3) Softmax1 only suppresses the contribution of tokens with lower weights, instead of pruning it from the computational graph, thus there may not be actual reduction of computational overhead.

(4) The method requires supervised finetuning to replace softmax with softmax1 in pretrained models. This could be relatively impractical for large-scale experiments, and may also affect the generalizability of pretrained models (due to finetuning on specific datasets).

**Questions:**

(1) Please justify the technical contribution of the proposed studies over prior ones using softmax1 activation.

(2) How does the observation in Section 3 motivate the use of softmax1?

(3) How does incorporating softmax1 affect the attention behaviors across layers for both inference and during training?

(4) How would the proposed method reduce the computational overhead in practice?

(5) While I am aware of the focus on mathematical reasoning in this paper, the requirements of supervised fine-tuning makes me concerned about the potential issues on model generalization. For instance, does fine-tuning on mathematical dataset hurt the model performance on other reasoning tasks? How would training models with softmax1 from scratch work, compared to post-hoc fine-tuning?

---

> ### Author Response · Authors · 2025-11-17
> **Rebuttal**
>
> > **Reviewer's Comment**: Prior studies ....
>
> **Response**: We appreciate the reviewer’s insightful comparison with [1]. While [1] indeed explored the use of Softmax₁ for attention sink in general LLM optimization, our work fundamentally differs in scope, granularity, and objective. Specifically, FROST extends Softmax₁ from token-level activation control to sentence-level reasoning outlier removal, a new concept first defined in our paper (**§3.3**). This shift enables pruning of entire uncritical reasoning sentences rather than isolated tokens, which is essential for efficient reasoning. Moreover, we clarify that the visualization technique used in **Figure 2** follows a common practice in attention analysis (not originally proposed in [1]). Despite using a similar visualization tool, our observations are distinct: [1] highlights early-layer attention sinks on initial tokens, whereas our results reveal sentence-level reasoning outliers on non-critical reasoning traces. In contrast to [1]’s goal of avoiding early-token collapse, FROST leverages Softmax₁ to suppress full reasoning outliers—thereby addressing efficiency and reliability at a higher linguistic unit. We have provided a complete theoretical proof that Softmax₁ can suppress sentence-level attention outliers in **section 5**.
>
> > **Reviewer's Comment**: The paper highlights ....
>
> **Response**: We appreciate the reviewer’s detailed question and agree that connecting our method to the observations in **Section 3** is important. We analyzed the effect of Softmax₁ across transformer layers by visualizing the attention distributions of head 15 for both vanilla Softmax and Softmax₁. As shown in **Figures 7 and 8 in the Appendix I**, Softmax₁ consistently suppresses attention outliers, resulting in smoother and more stable activations throughout the network. In lower layers, Softmax₁ contracts heavy tails and mitigates rare extreme peaks, thereby enhancing local feature mixing with higher-entropy and reduced-kurtosis distributions. In higher layers, it suppresses residual long-range spikes and sharpens attention toward semantically relevant tokens, producing sparser yet more stable focus and clearer causal information flow. These empirical findings confirm that the theoretical progressive-refinement behavior analyzed in **Figure 4** holds in practice—Softmax₁ enhances, rather than diminishes, the refinement dynamics of reasoning across layers. Also,  Softmax₁ primarily prunes tokens with low attention weights and negligible contributions to the final answer. These tokens typically correspond to redundant verification, self-checking, or repetitive reasoning traces that inflate token usage without improving reasoning quality. By contracting the heavy tails of the attention distribution, Softmax₁ suppresses such low-importance tokens while preserving and amplifying attention on semantically critical reasoning tokens that drive causal inference. This selective pruning enables the model to focus on meaningful reasoning paths and reduces computational overhead. We provide a detailed real example in **Appendix E and Appendix K** presents a newly added human-expert verification experiment, which shows FROST achieves **92%** accuracy in correctly removing non-critical reasoning traces.
>
>
> > **Reviewer's Comment**: Softmax1 only suppresses ...
>
> **Response**: We appreciate the reviewer’s insightful observation. While Softmax₁ does not explicitly prune tokens from the computational graph, it performs functional pruning through probability suppression, consistent with the outlier-efficient mechanism analyzed in GERM[2]. When attention weights decrease, Softmax₁ sharpens the distribution such that low-value sentences are effectively driven toward zero probability. This makes these sentences nearly inactive in the attention aggregation and drastically lowers their chance of being autoregressively generated. Although the decoding process remains token-level, the suppressed sentences have vanishing generation likelihood, thereby reducing total produced tokens and inference cost. Empirically, this probabilistic suppression leads to a **58.72 %** reduction in token usage in **Table 1** and at least **28.6%** decrease in inference time in **Table 4**, demonstrating that Softmax₁ not only mitigates reasoning outliers but also achieves tangible computational savings in practice.
>
> [1] Xiao, Guangxuan, Yuandong Tian, Beidi Chen, Song Han, and Mike Lewis. "Efficient streaming language models with attention sinks." arXiv preprint arXiv:2309.17453 (2023).
>
> [2] Luo, Haozheng, Chenghao Qiu, Maojiang Su, Zhihan Zhou, Zoe Mehta, Guo Ye, Jerry Yao-Chieh Hu, and Han Liu. "Fast and Low-Cost Genomic Foundation Models via Outlier Removal." In Forty-second International Conference on Machine Learning.

---

> > ### Author Response · Authors · 2025-11-17
> > **Rebuttal - Continue**
> >
> > > **Reviewer's Comment**: The method requires supervised finetuning to replace softmax with softmax1 in pretrained models. This could be relatively impractical for large-scale experiments, and may also affect the generalizability of pretrained models (due to finetuning on specific datasets).
> >
> > **Response**: We appreciate the reviewer’s insightful question. As shown in **Table 4**, the finetuning required for Softmax₁ is lightweight and substantially more efficient than compared approaches. In terms of training cost, our method is nearly **1.75x** faster than DRP (another SFT-based approach) and **5.8x** faster than ThinkLess (RL-based). At test time, it is **6x** faster than DRP and also faster than ThinkLess. Therefore, although our method involves a small amount of supervised finetuning, its computational cost is significantly lower than comparable baselines, making it practical for large-scale use.
> > Regarding generalizability, our finetuning is conducted on OpenR1—a broad, non–task-specific reasoning corpus—rather than on any benchmark-specific data. As shown in the paper **Table 1**, the model is evaluated on four out-of-domain mathematical reasoning datasets (GSM8K, MATH500, AIME24, Minerva), all distributionally distinct from OpenR1. The consistently strong results on these OOD benchmarks confirm that our lightweight finetuning does not harm generalizability and instead strengthens reasoning robustness.
> >
> > > **Reviewer's Comment**: While I am aware of the focus on ...
> >
> > **Response**: We thank the reviewer for raising these practical and generalization concerns. As illustrated in **Figure 5**, FROST replaces the vanilla Softmax with Softmax₁ and performs a brief supervised fine-tuning (SFT) stage to adapt parameters to the new activation. Unlike prior works that required full retraining or multi-stage continual learning, FROST achieves effective reasoning outlier removal with only a few fine-tuning steps from existing pretrained checkpoints using LoRA adapters (**rank = 8**). This lightweight process limits computational cost and preserves the pretrained backbone, making the approach practical for large-scale models.
> >
> > Regarding generalization, our experiments across two reasoning **models—Phi-4-Reasoning and GPT-OSS-20B**—and **four** math benchmarks demonstrate that Softmax₁ consistently improves accuracy and efficiency without degrading performance on out-of-domain math reasoning tasks. Similar to findings in the GERM[2] and SPARQ[3] studies, training models with Softmax₁ from scratch should achieve **at least equivalent or better** results than post-hoc fine-tuning, as it eliminates the activation-mismatch phase. However, we emphasize that our few-step LoRA-based SFT is the most computationally feasible and performance-preserving route for large reasoning models, maintaining general reasoning ability while improving efficiency. To further demonstrate the generalizability of FROST, we conducted an additional experiment comparing its reasoning performance on several basic tasks beyond mathematical reasoning. We evaluated FROST on **code generation and physics** tasks using the same Phi-4-Reasoning setup in **Table5** and found that it not only preserves but improves performance on these out-of-domain tasks. This is expected because FROST removes uncritical reasoning traces in a way that naturally transfers beyond the finetuning domain. Also, since FROST only swaps the attention activation for Softmax₁ and applies lightweight LoRA updates, the parameter changes are minimal, allowing the model’s overall reasoning generalization to remain intact.
> >
> > | Method | Leetcode Pass@1 | Leetcode #Tk | LiveCodeBench Pass@1 | LiveCodeBench #Tk | UGPhysics Pass@1 | UGPhysics #Tk | $\overline{\text{Pass@1}}$ | Avg #Tk|
> > |:---:|:---:|:---:|:---:|:---:|:---:|:---:|:---:|:---:|
> > | Base | 0.3222 | 2755.13 | 0.3248 | 3154.80 | _0.3172_ | _2603.00_ | 0.3214 | 2837.64 |
> > | Softmax | _0.3778_ | _2106.85_ | _0.3538_ | _2909.07_ | 0.3011 | 2622.52 | _0.3442_ | _2546.15_ |
> > | **FROST** | **0.3889** | **1163.06** | **0.3777** | **1967.56** | **0.3473** | **805.77** | **0.3713** | **1312.13** |
> >
> > [3] Wu, Shang, et al. "Fast adaptation and robust quantization of multi-modal foundation models from associative memory: A case study in speechLM." Workshop on Efficient tems for Foundation Models II@ ICML2024. 2024.

---

> ### Author Response · Authors · 2025-11-17
> **Rebuttal - Continue**
>
> > **Reviewer's Comment**: Please justify the technical contribution of the proposed studies over prior ones using softmax1 activation.
>
> **Response**: We thank the reviewer for raising this important point. Prior studies [1] employ Softmax₁ primarily for token-level activation smoothing to stabilize gradients and mitigate early-token attention sinks. In contrast, our work introduces a new conceptual and algorithmic layer—sentence-level reasoning outlier removal—which redefines how Softmax₁ is applied. Specifically:
> New outlier concept: We introduce the concept of reasoning outliers and propose FROST to prune uncritical reasoning steps characterized by low attention.
> New unit of analysis: We extend Softmax₁ from token-level sparsification to sentence-level reasoning aggregation, enabling the model to identify and suppress entire uncritical reasoning sentences rather than individual tokens **(§3.3, §5)**.
> New theoretical guarantee: We derive Theorems **5.1–5.2**, proving that Softmax₁ enforces tail contraction and deployment-time suppression at the sentence granularity, establishing a formal connection between attention dynamics and reasoning efficiency.
> New empirical domain: We apply this mechanism to efficient reasoning, not generic LLM training. FROST demonstrates that sentence-level suppression yields shorter and more reliable reasoning trajectories—reducing token usage by **58.72%** and improving accuracy by **10.64%**—a distinct objective absent in prior Softmax₁ works.
>
> Hence, FROST transforms Softmax₁ from a low-level activation trick into a reasoning-oriented outlier removal framework with new theory, scope, and efficiency results beyond previous studies
>
> > **Reviewer's Comment**: How does the observation in Section 3 motivate the use of softmax1?
>
> **Response**: We appreciate the reviewer’s insightful question. The analyses in Section 3 reveal that large reasoning models exhibit **uneven attention distributions** across reasoning steps—early and verification sentences receive low attention, while critical reasoning traces concentrate higher attention mass. This progressive refinement indicates that most inefficiency arises from *reasoning outliers*—sentences with persistently low attention and negligible contribution to the final answer.
> Motivated by this empirical pattern, we adopt **Softmax₁** because its *tail-contraction property* (**Theorem 5.1**) probably suppresses these low-attention regions while preserving high-attention ones, thereby amplifying the model’s natural refinement process. In effect, Softmax₁ operationalizes the **Section 3** finding—turning the observed attention sparsity into a controllable mechanism that prunes reasoning outliers at the sentence level without damaging critical reasoning continuity.
>
> > **Reviewer's Comment**: How does incorporating softmax1 affect the attention behaviors across layers for both inference and during training?
>
> **Response**: We appreciate the reviewer’s insightful question.  Integrating Softmax₁ reshapes layer-wise attention to eliminate reasoning outliers and accelerate training convergence. During training, Softmax₁ performs tail contraction, suppressing noisy low-attention activations and thereby reducing gradient variance. This leads to faster and more stable optimization—particularly under LoRA-based SFT, where Softmax₁ enables the low-rank adapters to more rapidly cover the effective attention subspace. This observation is consistent with GERM[2]. During inference, the same mechanism sharpens attention in deeper layers, reducing redundant reasoning sentences and reinforcing cross-layer focus on high-value reasoning traces (as shown by the lower kurtosis and higher sentence entropy in **Table 3**. Overall, Softmax₁ not only enhances inference efficiency but also accelerates LoRA SFT convergence by improving attention signal quality. We have addeda detailed discussion of these cross-layer and convergence effects in **Appendix H**.
>
> > **Reviewer's Comment**: How would the proposed method reduce the computational overhead in practice?
>
> **Response**: We appreciate the reviewer’s insightful question. By replacing Softmax with Softmax₁, FROST contracts the attention tail so low-weight reasoning outlier sentences receive near-zero probability mass; this makes those sentences unlikely to be generated, shortening trajectories and cutting KV-cache growth and attention/FFN calls per sample. Empirically, this yields large end-to-end savings—**58.72%** fewer tokens on average and faster wall-clock inference (**Table 4**)—demonstrating practical compute reduction despite unchanged per-step FLOPs.

---

> > ### Author Response · Authors · 2025-11-25
> >
> > Dear Reviewer,
> >
> > We sincerely appreciate your thoughtful feedback. As the discussion period reaches its final week, we would be grateful to know whether our latest responses have resolved your concerns or if any questions remain. We are happy to provide further clarification as needed. Wishing you a Happy Thanksgiving.
> >
> > Thanks,
> >
> > Authors

---

> > ### Comment · Reviewer_ncQk · 2025-11-27
> >
> > I thank the authors for the clarification and additional experiments. The new results address most of my concerns, e.g., the effects of softmax1 on suppressing non-critical tokens, generalizability to non-mathematical problems, and technical contributions.
> >
> > Regarding the computational overhead. As the authors mention that the reduction is achieved by generating more efficient responses, I would suggest a more direct comparison on length of generated answers between the original and the proposed method.
> >
> > I will adjust my rating upon reading other reviewers' responses, but I am leaning towards raising the score.

---

> > > ### Author Response · Authors · 2025-11-27
> > >
> > > We appreciate the reviewer’s constructive suggestion regarding response length. As noted, our evaluation already reports the number of generated tokens in **Table 1,2,3,5**, and these results were included in the initial draft. Since autoregressive LRMs produce text strictly token-by-token, this metric directly reflects the actual response length and is the standard measure used in prior efficient-reasoning studies [1-5].
> > >
> > > Across models and datasets, FROST achieves a **58.72% average reduction** in token count, demonstrating a substantial decrease in response length while simultaneously improving accuracy. This percentage reduction directly captures the computational savings produced by pruning non-critical reasoning steps.
> > >
> > > If any further clarification would be helpful, we are happy to provide more detail. We sincerely appreciate your positive inclination and hope these strengthened results support a favorable final score.
> > >
> > > [1] Jiang, Y., Li, D., & Ferraro, F. (2025). DRP: Distilled Reasoning Pruning with Skill-aware Step Decomposition for Efficient Large Reasoning Models. arXiv preprint arXiv:2505.13975.
> > >
> > > [2] Li, G., Gao, Y., Li, Y., & Wu, Y. (2025). ThinkLess: A Training-Free Inference-Efficient Method for Reducing Reasoning Redundancy. arXiv preprint arXiv:2505.15684.
> > >
> > > [3] Han, T., Wang, Z., Fang, C., Zhao, S., Ma, S., & Chen, Z. (2025, July). Token-budget-aware llm reasoning. In Findings of the Association for Computational Linguistics: ACL 2025 (pp. 24842-24855).
> > >
> > > [4] Ma, X., Wan, G., Yu, R., Fang, G., & Wang, X. (2025). Cot-valve: Length-compressible chain-of-thought tuning. arXiv preprint arXiv:2502.09601.
> > >
> > > [5] Xia, Heming, et al. "Tokenskip: Controllable chain-of-thought compression in llms." arXiv preprint arXiv:2502.12067 (2025).

---

### Official Review · Reviewer_vRQV · 2025-11-01

**Soundness:** 3
**Presentation:** 3
**Contribution:** 3
**Rating:** 6
**Confidence:** 3

**Summary:**

LRMs typically generate a large number of noncritical steps, inevitably introducing inefficiencies and potential inaccuracies. The authors propose FROST, a method for accelerating and strengthening inference. They first introduce the concept of inference outliers: noncritical steps with low attention weights and low entropy. Technically, the authors use softmax1 to detect and eliminate inference outliers. Preliminary experiments effectively reveal the problem FROST aims to address, and comprehensive experiments demonstrate its effectiveness.

**Strengths:**

1.The authors clearly articulate and demonstrate the problem of outliers in reasoning through premise experiments, while also providing a simple yet highly effective solution. This is easy to understand and well-intentioned.
2.The experimental results on multiple datasets are significant and compared with the state-of-the-art algorithms, convincingly demonstrating the effectiveness of FROST.
3.In addition to basic performance experiments, the paper also includes a large number of validation experiments, such as using some metrics to try to quantify the performance of outlier removal.

**Weaknesses:**

1.In Figure 4, the attention weight of S2 is also significantly compressed, while S3 is actually improved. I understand that S24 is the key reasoning path and has been improved. However, further analysis of S2 and S3 is necessary.
2.The authors demonstrate in Table 3 that removing attention outliers increases the probability assigned to critical sentences. Token entropy is used as an indicator of criticality. While most experiments meet expectations, Entmax15 and Sparsemax exhibit unexpected performance. The authors offer no explanation. This is necessary to clarify whether removing inference outliers actually improves reasoning ability. Please try to provide possible explanations as to whether there is token entropy or some other fluctuation.

**Questions:**

1.The x and y axes of all the graphs are unclear, especially in graph 2, where the y-axis of the two upper subgraphs is even all zero due to missing parts. Could you please make the graphs clearer?

---

> ### Author Response · Authors · 2025-11-17
>
> > **Reviewer's Comment**: In Figure 4, the attention weight of ...
>
> **Response**: We appreciate the reviewer’s observation and agree that a deeper explanation of **S2 and S3** helps clarify the results in **Figure 4**. As detailed in **Appendix E**, we analyzed the Phi-4-Reasoning model’s response to the first question of AIME24. Traces **S1 and S2** are classified as uncritical reasoning steps. Although **S2** includes partially meaningful content (e.g., “So the walking time (actual walking time) plus t minutes equals total time.”), The overall reasoning is dominated by redundant verification and restatement, leading Softmax₁ to compress its attention weight. In contrast, **S3** represents a critical reasoning step where the model identifies the two key governing equations, so its attention weight is reinforced. From **S4 to S19**, the model enters a self-verification loop (e.g., traces beginning with “wait”), which FROST effectively suppresses. Starting from **S20**, the model resumes coherent reasoning and completes the solution at **S24**. This analysis confirms that Softmax₁ selectively down-weights partially redundant sentences like **S2** while amplifying genuinely causal reasoning traces such as **S3 and S24**.
>
> > **Reviewer's Comment**: The authors demonstrate in Table 3 that removing attention outliers ....
>
> **Response**: We thank the reviewer for highlighting this point. The discrepancy in Entmax15 and Sparsemax performance arises from their inherent sharpening behavior—both suppress not only low-value but also high-value attention scores, which can truncate parts of critical reasoning traces. As a result, while the attention outlier metrics (maximum infinity norm and kurtosis) measure internal activation distributions inside the model, the token entropy and Pass@1 reflect external output behaviors. Because both Entmax15 and Sparsemax reshape the attention in a similar manner, they produce comparable output entropy and Pass@1 scores, even though their internal activation patterns differ. Entmax15, being smoother, mitigates this effect slightly, whereas Sparsemax shows stronger truncation and larger loss in reasoning coverage. We will clarify this distinction between internal outlier metrics and external output evaluations in the revision to emphasize how Softmax₁ preserves reasoning entropy while maintaining internal stability.
>
> > **Reviewer's Comment**: The x and y axes of all the graphs are unclear, ....
>
> **Response**: We thank the reviewer for pointing out the clarity issue in **Figure 2**. We have already corrected the plotting script to ensure all axes are properly labeled and scaled. The zeroed y-axis values in the upper plots were caused by a typo during figure organization, which has been fixed in the revised version. We will include the updated, high-resolution figures with clear axis annotations and consistent scales across subplots to enhance readability and interpretability.

---

> > ### Author Response · Authors · 2025-11-25
> >
> > Dear Reviewer,
> >
> > We sincerely appreciate your thoughtful feedback. As the discussion period reaches its final week, we would be grateful to know whether our latest responses have resolved your concerns or if any questions remain. We are happy to provide further clarification as needed. Wishing you a Happy Thanksgiving.
> >
> > Thanks,
> >
> > Authors

---

> > > ### Author Response · Authors · 2025-11-28
> > >
> > > Dear reviewers,
> > >
> > > As the rebuttal deadline is approaching, I would like to kindly check whether our responses have addressed your concerns and questions. If there is anything that is still unclear or could benefit from further clarification, please let us know, and we will be happy to elaborate.
> > >
> > > Thank you very much for your time and effort in reviewing our work.
> > >
> > > Happy Thanksgiving!

---

### Author Response · Authors · 2025-11-17
**General Rebuttal/Revision Response**

We sincerely thank the reviewers for their insightful feedback and constructive suggestions, which have greatly improved the technical rigor and clarity of our work.

**This paper presents an efficient reasoning framework that leverages attention dynamics to prune uncritical reasoning paths (Sec. 3 & 4**). Below, we summarize the key strengthened contributions and major revisions incorporated during the rebuttal.

---
## Key Contributions:
- **New outlier concept:** We introduce the concept of reasoning outliers and propose FROST to prune uncritical reasoning steps characterized by low attention.
- **New unit of analysis:** We extend Softmax₁ from token-level sparsification to sentence-level reasoning aggregation, enabling the model to identify and suppress entire uncritical reasoning sentences rather than individual tokens **(§3.3, §5)**.
- **New theoretical guarantee:**  We derive Theorems **5.1–5.2**, proving that Softmax₁ enforces tail contraction and deployment-time suppression at the sentence granularity, establishing a formal connection between attention dynamics and reasoning efficiency.
- **New empirical domain:** We apply this mechanism to efficient reasoning, not generic LLM training. FROST demonstrates that sentence-level suppression yields shorter and more reliable reasoning trajectories—reducing token usage by **58.72%** and improving accuracy by **10.64%**—a distinct objective absent in prior Softmax₁ works.

---
## Major Revisions:

- **Contribution Clarification**  `Reviewer  ncQk and ej9v`
- **New Appendix I**: Added detailed analysis of Softmax₁ behavior during training and inference. `Reviewer  ncQk and ej9v`
- **Revised Sec. 6.1**: Explained the unexpected performance of Sparsemax and Entmax variants. `Reviewer vRQV`
- **New Experiments**:
    - **New Sec. 6.2 & Table 5**: Added experiments on model generalizability after applying FROST. `Reviewer  ncQk`
    - **New Appendix K**: Included results from human evaluation experiments. `All reviewers`
    - **New Appendix H & Figure 7**: Added new experiments analyzing the impact of Softmax₁ across different Transformer layers. `Reviewer  ncQk`
    - **New Appendix J & Figure 9**: Introduced additional visualization experiments showing attention heatmaps across various layers and heads. `Reviewer oTRF`
    - **New Figure 8**: Presented further experiments illustrating the influence of Softmax₁ in other layers to validate layer-wise consistency.  `Reviewer  ncQk`
    - **New Appendix E**: Added a real-world example illustrating LRM reasoning traces, demonstrating how the model performs multi-step reasoning in a practical scenario. `All reviewers`
  - **Revised Table 1**: An additional experiment using Magistral-Small-1.1 as an extra LRM evaluation. `Reviewer  oTRF`
- **Revised Sec. 2**: Expanded the related work discussion to include the contributions of Choi et al. and Cai et al., providing broader context on prior approaches relevant to our study. `Reviewer oTRF`

## Minor Revisions
- **Clarified Outlier Metrics**: Explained why the maximum infinity norm and average kurtosis are used as quantitative indicators of outlier severity—capturing, respectively, the magnitude of extreme activations and the overall heaviness of activation tails. `Reviewer  ej9v`
- **Corrected Figures 2 & 3**: Fixed labeling inconsistencies and ensured that axis titles and legends accurately reflect the experimental variables. `Reviewer vRQV`
- **Revised abstract, Sec 1, Sec 6.1, Sec 7**: Updated performance improvements after incorporating the new LRM experiment.  `Reviewer oTRF`
---
We once again thank the Area Chair and reviewers for their valuable efforts and thoughtful comments. All revisions have been incorporated into the updated manuscript, with changes highlighted in blue.

---

### Author Response · Authors · 2025-11-23
**Friendly Reminder**

Dear Reviewers,

We sincerely appreciate your thoughtful feedback. As the discussion period reaches its midpoint, we would be grateful to know whether our latest responses have resolved your concerns or if any questions remain. We are happy to provide further clarification as needed. Wishing you a Happy Thanksgiving.

Thanks,

Authors

---

### Author Response · Authors · 2025-12-01
**Global Summarize**

Dear Chairs,


We sincerely thank the area chair and all reviewers for the time and effort dedicated to evaluating our submission.


The paper initially received four reviews with scores of **2, 6, 6, 6**, and after we addressed all questions and concerns during the rebuttal phase, the ratings improved to **[Intent to Increase], 6, 6, 6**, with all finalizations occurring **at least one day before the information leak.**


Below, we describe how [our rebuttal](https://openreview.net/forum?id=a9dngZLqGS&noteId=dgGIlCy7N3) contributed to these outcomes.


---


## Reviewer [ncQk](https://openreview.net/forum?id=a9dngZLqGS&noteId=yTZujXCpKo) (Rating: 2 → leaning toward increase, Confidence: 3)

**Initial concerns:**

- Limited novelty: Softmax₁ previously studied; unclear technical contribution.
- Disconnect between observations (**Section 3**) and proposed method.
- Practicality: supervised fine-tuning may harm generalization and increase cost.
- Lack of clarity on how Softmax₁ affects attention across layers and reduces overhead.
- The ability of the method to generalize to non-mathematical reasoning tasks.

**Rebuttal actions:**

- **Clarified novelty:** FROST introduces sentence-level reasoning outlier removal, new theoretical guarantees **(Theorems 5.1–5.2)**, and efficiency objectives absent in prior work.
- Connected Section 3 observations to Softmax₁’s tail-contraction property for pruning low-attention sentences.
- **Added Appendix I and H:** visualizations showing layer-wise suppression of outliers and refinement dynamics.
- **Demonstrated practical efficiency:** 58.72% token reduction, 28.6% faster inference, lightweight LoRA-based fine-tuning **(1.75× faster than DRP, 5.8× faster than ThinkLess).**
- **Added generalization experiments on code and physics tasks (Table 5**), confirming no degradation beyond math reasoning.

**Reviewer response:**
> “I thank the authors for the clarification and additional experiments. The new results address most of my concerns, e.g., the effects of softmax1 on suppressing non-critical tokens, generalizability to non-mathematical problems, and technical contributions.
Regarding the computational overhead. As the authors mention that the reduction is achieved by generating more efficient responses, I would suggest a more direct comparison on length of generated answers between the original and the proposed method.
I will adjust my rating upon reading other reviewers' responses, but I am leaning towards raising the score."

We responded to `reviewer ncQk` and clarified that the number of tokens is an important metric for measuring response length. However, due to the timing of the incident, the reviewer did not have the opportunity to reply.


## Reviewer [oTRF](https://openreview.net/forum?id=a9dngZLqGS&noteId=cgrm5b839b) (Rating: 6 → 6, Confidence: 3)

**Initial concerns:**

- Restricted experimental scope (mostly Phi-4-Reasoning).
- Limited related work discussion; novelty unclear relative to token/KV compression methods.
- Visualization choices (layers/heads) lacked justification.

**Rebuttal actions:**
- Added experiments on **Magistral-Small-1.1**, showing consistent gains **(26.7% accuracy improvement, 69.68% token reduction across models).**
- Expanded related work section contrasting FROST’s sentence-level pruning with token/KV compression approaches.
- Justified visualization choices and added extended visualizations in **Appendix J.**

**Reviewer response:**

> “Thank you for response. For the revision, it would be important to include extended experiments on additional reasoning models (e.g., R1-distill) and to provide a more detailed related work section.“

We responded to `reviewer oTRF` with new experiment on **Magistral-Small-1.1**. However, due to the timing of the incident, the reviewer did not have the opportunity to reply.

---
Below, we explain how our rebuttal addressed reviewers `n6JN`’s (**rating 6**) and `ej9v`’s (**rating 6**) concerns. Neither not responded yet.

## Reviewer [ej9v](https://openreview.net/forum?id=a9dngZLqGS&noteId=rfy1FCqswr) (Rating: 6 → 6, Confidence: 2, not response yet)

**Initial concerns:**
- Limited novelty: Softmax₁ known for suppressing low-value attention.
- Implicit detection of reasoning outliers; framing feels post-hoc.
- Choice of outlier metrics (‖x‖∞, kurtosis) and qualitative pruning examples.

**Rebuttal actions:**
- Explicitly acknowledged prior work and clarified novelty: new concept of reasoning outliers, sentence-level pruning, theoretical guarantees, and efficiency gains.
- Explained Softmax₁ as a continuous outlier detector enabling dynamic suppression without heuristics.
- Justified metrics as standard indicators of activation outliers;
- Added qualitative examples and human annotation study (**92% pruned steps = redundant self-corrections**).

---

> ### Author Response · Authors · 2025-12-01
> **Global Summarize - Continued**
>
> ## Reviewer [vRQV](https://openreview.net/forum?id=a9dngZLqGS&noteId=WYe7dhYhvv) (Rating: 6, Confidence: 3, not response yet)
>
> **Initial concerns:**
> - Needed deeper analysis of attention behavior (e.g., S2 vs S3 in Figure 4).
> - Unexpected performance of Entmax/Sparsemax variants; unclear explanation.
> - Graph clarity issues.
>
> **Rebuttal actions:**
> - Added detailed analysis of reasoning traces (**Appendix E**) explaining selective suppression of redundant steps.
> - Clarified why Entmax/Sparsemax truncate critical reasoning traces due to sharpening behavior.
> - Fixed figure labeling and axis issues; provided high-resolution plots.
>
> ---
>
>
> ## Notes on `Reviewer ncQk`’s Score Adjustment
>
>
> However, we still have concerns about `Reviewer ncQk`'s evaluation because the reasons for the low score from the `Reviewer ncQk` are:
> - A limited understanding or awareness of how outlier-induced effects distort the reasoning trace.
> - A confusion between “sentence-level outlier removal” and “token-level attention sinking”, which are fundamentally different phenomena.
> - The confusion comes from overlooking that token count is a common metric for response length.
>
> Given these points, we kindly invite the Chairs to review these clarifications as well, to help ensure a balanced and accurate assessment. We acknowledge that some concerns may have arisen from misunderstandings, and reviewer ncQk had already indicated recognition of the clarification regarding our contributions. Due to the incident, the corresponding score update is not reflected on our paper. We therefore hope the chairs can take this into account to ensure a balanced and accurate assessment.
>
> ---
>
>
> Thank you for your consideration.
>
>
> Authors

---

### Meta-Review · Area_Chair_JCwH · 2026-01-05

**Summary:**

This paper proposes FROST, a method that replaces standard Softmax with Softmax1 in attention layers to suppress "reasoning outliers" (uncritical reasoning steps with low attention weights).

Reviewers raised concerns about: (1) limited novelty given prior work on Softmax1 for attention sinks, (2) restricted experimental scope (primarily Phi-4-Reasoning), (3) disconnect between Section 3 observations and the proposed method, (4) generalization beyond mathematical reasoning, and (5) clarity of figures and explanations.

**Reviewer Concerns:**

- Generalization concerns were addressed through new experiments on code/physics tasks and an additional model (Magistral-Small)
- The novelty distinction (sentence-level vs. token-level pruning) was clarified with new theoretical framing
- Figure clarity issues were corrected
- Layer-wise visualizations were added


- The core novelty argument remains somewhat incremental: applying a known technique to a new domain
- Related work section still needs expansion comparing to KV compression methods

**Reviewer Scores:**

ncQk: 2 to likely 5-6 (explicitly stated "leaning towards raising the score" after rebuttal)
oTRF: 6 to 6 (satisfied with additional experiments but requested more models)
ej9v: 6 to 6 (no response, concerns partially addressed)
vRQV: 6 to 6 (no response, concerns addressed)

---

### Decision · Program_Chairs · 2026-01-26

Accept (Poster)